# False Discovery Proportion control for aggregated Knockoffs

**Alexandre Blain**
INRIA
Université Paris-Saclay
alexandre.blain@inria.fr

**Bertrand Thirion**
INRIA
CEA
bertrand.thirion@inria.fr

**Olivier Grisel**
INRIA
olivier.grisel@inria.fr

**Pierre Neuvial**
Institut de Mathématiques de Toulouse
Université de Toulouse
pierre.neuvial@math.univ-toulouse.fr

## Abstract

Controlled variable selection is an important analytical step in various scientific fields, such as brain imaging or genomics. In these high-dimensional data settings, considering too many variables leads to poor models and high costs, hence the need for statistical guarantees on false positives. Knockoffs are a popular statistical tool for conditional variable selection in high dimension. However, they control for the expected proportion of false discoveries (FDR) and not their actual proportion (FDP). We present a new method, KOPI, that controls the proportion of false discoveries for Knockoff-based inference. The proposed method also relies on a new type of aggregation to address the undesirable randomness associated with classical Knockoff inference. We demonstrate FDP control and substantial power gains over existing Knockoff-based methods in various simulation settings and achieve good sensitivity/specificity tradeoffs on brain imaging and genomic data.

## 1 Introduction

Statistically controlled variable selection arises in many different application fields, when the aim is to identify variables that are important for predicting an outcome of interest. For instance, in the context of brain imaging, practitioners are interested in finding which brain areas are relevant for predicting behavior or brain diseases. Such problems also appear in genomics, where practitioners wish to select genes associated with disease outcomes.

More precisely, we consider here *conditional* variable selection, meaning that we wish to select variables that are relevant to predict an outcome *given* the other variables. This type of inference is substantially more challenging than marginal inference, especially in high-dimensional settings, where the number of variables exceeds the number of samples. This is typically the case for brain mapping studies that comprise at most a few hundred subjects (hence, samples), while modern functional Magnetic Resonance Imaging (MRI) scans consist of more than 100k *voxels*. In the context of conditional inference, those are typically reduced to a few hundreds of brain regions, still possibly more than the number of samples.

Importantly, statistical guarantees are needed to ensure that the inference is reliable - i.e. that the proportion of false discoveries made by the variable selection procedure is controlled.

In the Knockoffs framework [1, 6], this problem is tackled by building noisy copies of the original variables. These copies are then compared to their original counterpart to perform variable selection.

37th Conference on Neural Information Processing Systems (NeurIPS 2023).

The intuition underlying Knockoffs is that irrelevant variables do not get a larger weight than their Knockoff, while relevant variables do. Crucially, the Model-X Knockoffs procedure [6] controls the False Discovery Rate [2] which is the expected proportion of false discoveries.

A major caveat with this procedure is the random nature of the Knockoffs generation process: for two runs of the Knockoffs procedure on the same data, different Knockoffs will be built and subsequently different variables may be selected. This undesirable behavior hinders reproducibility. A second caveat is that False Discovery Rate (FDR) control does not imply False Discovery Proportion (FDP) control [12]. This leads to potentially unreliable inference: single runs of the method can produce a much higher proportion of False Discoveries than the chosen FDR level.

In this work, we propose a novel Knockoff-based inference procedure that addresses both concerns while offering power gains over existing methods, for no significant computation cost. The paper is organized as follows. After a refresher on Knockoff inference and aggregation, we consider the $\pi$ statistic introduced in [17] to rank variables by relevance. Using the symmetry of knockoffs under the null hypothesis, we construct explicit upper bounds on the Joint Error Rate (JER; 4) of these statistics, leading to FDP control. We then use the calibration principle of [4] to obtain sharper bounds. Finally, we obtain a robust version of this method using harmonic mean aggregation of the $\pi$ statistics across multiple Knockoffs draws. We demonstrate empirical power gains in various simulation settings and show the practical benefits of the proposed method for conditionally important region identification on fMRI and genomic datasets.

## 2 Related work

There has been much effort in the statistical community to achieve derandomized Knockoff-based inference. [19] introduced the idea of running Model-X Knockoffs [6] multiple times and computing for each the proportion of runs for which it was selected. [9] explore the idea of sampling multiple Knockoffs simultaneously. This induces a massive computational cost, which is prohibitive compared to methods that can support parallel computing. [17] introduced an aggregation method that relies on viewing Model-X Knockoffs as a Benjamini-Hochberg (BH) procedure [2] on so-called *intermediate p-values*. Such $p$-values can be computed on different Knockoff runs and aggregated using quantile aggregation [15] – then, BH is performed on the aggregated $p$-values to select variables. This approach relies on the heavy assumption that Knockoff statistics are i.i.d. under the null. Additionally, it is penalized by the conservativeness of the quantile aggregation scheme. Alternative aggregation schemes such as the harmonic mean [28] can be used but do not yield valid $p$-values.

[18] introduced an alternative aggregation procedure where Model-X Knockoffs are viewed as an e-BH procedure [25] on well-defined e-values [24]. Since the mean of two e-values remains an e-value, aggregation is done by averaging e-values across different Knockoffs draws. Then, e-BH is performed on the aggregated e-values to select variables. FDR control on aggregated Knockoffs is achieved without any additional assumption compared to Model-X Knockoffs. However, this method requires the difficult setting of a hyperparameter related to the chosen risk level, which highly impacts power in practice. Other recent developments in Knockoffs include the conditional calibration framework of Luo et al. [14] which aims at improving the power of Knockoffs-based methods.

There have been a few attempts at controlling other type 1 errors than the FDR using Knockoffs. [11] achieves k-FWER control and proposes that FDP control can be obtained by using a procedure that leverages joint k-FWER control. Recently, [13] introduced such a procedure to reach FDP control based on the k-FWER control introduced in [11]. In summary, the KOPI approach is the first one that aims at controlling the FDP of knockoffs-based inference for any aggregation scheme, leading to both accurate FDP control and increased sensitivity.

## 3 Refresher on Knockoffs

**Notation.** We denote vectors by bold lowercase letters. A vector $\mathbf{x} = \{x_1, \ldots, x_p\}$ from which we removed the $j^{th}$ coordinate is denoted by $\mathbf{x}_{-j}$, i.e. $\mathbf{x} \setminus \{x_j\}$. Independence between two random vectors $\mathbf{x}$ and $\mathbf{y}$ is denoted by $\mathbf{x} \perp \mathbf{y}$. For two vectors $\mathbf{x}$ and $\tilde{\mathbf{x}}$ and a subset $S$ of indices, $(\mathbf{x}, \tilde{\mathbf{x}})_{swap(S)}$ denotes the vector obtained from $(\mathbf{x}, \tilde{\mathbf{x}})$ by swapping the entries $x_j$ and $\tilde{x}_j$ for each $j \in S$. Matrices are denoted by bold uppercase letters, the only exception being the vector of Knockoff statistics

that we denote by $\mathbf{W}$ as in [1, 6]. For any set $S$, $|S|$ denotes the cardinality of $S$. For a vector $\mathbf{z} = (z_j)_{1 \leq j \leq p}$ and $S \subset [\![p]\!]$, we denote by $z_{(j:S)}$ (or $z_{(j)}$ when there is no ambiguity) the $j^{th}$ smallest value in the sub-vector $(\mathbf{z}_s)_{s \in S}$. For an integer $k$, $[\![k]\!]$ denotes the set $\{1, \ldots, k\}$. Equality in distribution is denoted by $\overset{d}{=}$.

**Problem setup.** The input data are denoted by $\mathbf{X} \in \mathbb{R}^{n \times p}$, where $n$ is the number of samples and $p$ the number of variables. The outcome of interest is denoted by $\mathbf{y} \in \mathbb{R}^n$. The goal is to select variables that are relevant with regards to the outcome *conditionally on all others*. Formally, we test simultaneously for all $j \in [\![p]\!]$:

$$H_{0,j} : y \perp x_j | \mathbf{x}_{-j} \quad \text{versus} \quad H_{1,j} : y \not\perp x_j | \mathbf{x}_{-j}.$$

The output of a variable selection method is a rejection set $\hat{S} \subset [\![p]\!]$ that estimates the true unknown support $\mathcal{H}_1 = \{j : y \not\perp x_j | \mathbf{x}_{-j}\}$. Its complement is the set of true null hypotheses $\mathcal{H}_0 = \{j : y \perp x_j | \mathbf{x}_{-j}\}$. Its cardinality $|\mathcal{H}_0|$ is denoted by $p_0$. To ensure reliable inference, our aim is to provide a statistical guarantee on the proportion of False Discoveries in $\hat{S}$. The False Discovery Proportion (FDP) and the False Discovery Rate (FDR) [2] are defined as:

$$\text{FDP}(\hat{S}) = \frac{|\hat{S} \cap \mathcal{H}_0|}{|\hat{S}| \vee 1}, \quad \text{FDR}(\hat{S}) = \mathbb{E}[\text{FDP}(\hat{S})] = \mathbb{E}\left[\frac{|\hat{S} \cap \mathcal{H}_0|}{|\hat{S}| \vee 1}\right].$$

An $\alpha$-level post-hoc FDP upper bound [10] is a function $V$ that verifies:

$$\mathbb{P}\left(\forall S \subset [\![p]\!], \text{FDP}(S) \leq V(S)/|S|\right) \geq 1 - \alpha.$$

**Knockoffs.** The Knockoff filter is a variable selection technique introduced by [1] and refined by [6] which controls the FDR. This procedure relies on building noisy copies of the original variables called Knockoff variables, that are designed to serve as controls for variable selection.

**Definition 1** (Model-X Knockoffs, 6). For the family of random variables $\mathbf{x} = (x_1, \ldots, x_p)$, Knockoffs are a new family of random variables $\tilde{\mathbf{x}} = (\tilde{x}_1, \ldots, \tilde{x}_p)$ satisfying:

1. for any $S \subset [\![p]\!]$, $(\mathbf{x}, \tilde{\mathbf{x}})_{swap(S)} \overset{d}{=} (\mathbf{x}, \tilde{\mathbf{x}})$

2. $\tilde{\mathbf{x}} \perp \mathbf{y} | \mathbf{x}$.

Once we have such variables at our disposal, we quantify their importance relative to the original ones. This is done by computing Knockoff statistics $\mathbf{W} = (W_1, \ldots, W_p)$ that are defined as follows.

**Definition 2** (Knockoff Statistic, 6). A knockoff statistic $\mathbf{W} = (W_1, \ldots, W_p)$ is a measure of feature importance that satisfies:

1. $\mathbf{W}$ depends only on $\mathbf{X}, \tilde{\mathbf{X}}$ and $\mathbf{y}$: $\mathbf{W} = g(\mathbf{X}, \tilde{\mathbf{X}}, \mathbf{y})$.

2. Swapping column $\mathbf{x}_j$ and its knockoff column $\tilde{\mathbf{x}}_j$ switches the sign of $W_j$:

$$W_j([\mathbf{X}, \tilde{\mathbf{X}}]_{swap(S)}, \mathbf{y}) = \begin{cases} W_j([\mathbf{X}, \tilde{\mathbf{X}}], \mathbf{y}) \text{ if } j \in S^c \\ -W_j([\mathbf{X}, \tilde{\mathbf{X}}], \mathbf{y}) \text{ if } j \in S. \end{cases}$$

The most commonly used Knockoff statistic is the Lasso-coefficient difference (LCD) [27]. This statistic is obtained by fitting a Lasso estimator [22] on $[\mathbf{X}, \tilde{\mathbf{X}}] \in \mathbb{R}^{n \times 2p}$, which yields $\widehat{\boldsymbol{\beta}} \in \mathbb{R}^{2p}$. Then, the Knockoff statistic can be computed using $\widehat{\boldsymbol{\beta}}$:

$$\forall j \in [\![p]\!], \quad W_j = |\widehat{\beta}_j| - |\widehat{\beta}_{j+p}|.$$

This coefficient summarizes the importance of the original $j^{th}$ variable relative to its own Knockoff: $W_j > 0$ indicates that the original variable is more more important for fitting $y$ than the Knockoff variable, meaning that the $j^{th}$ variable is likely relevant. Conversely, $W_j < 0$ indicates that the $j^{th}$ variable is probably irrelevant. We thus wish to select variables corresponding to large and positive $W_j$. Formally, the rejection set $\hat{S}$ can be written $\hat{S} = \{j : W_j > T_q\}$, where $T_q$ is chosen to provably control the FDR at level $q$ [6].

**Aggregation schemes.** Due to the randomness in the knockoff generation process, different variables may be selected for two different runs of the method, which is undesirable. To mitigate this, aggregation of multiple Knockoffs runs is needed. Ren and Barber [18] introduced an aggregation scheme which relies defining Knockoffs $e$-values.

$$e_j = \frac{p}{1 + |\{k : W_k \leq -T_q\}|} 1_{\{W_j \geq T_q\}}.$$

Such e-values can be averaged across $D$ draws and e-BH [25] is performed for variable selection. Alternatively, [17] defines the following $\pi$-statistic, that quantifies the evidence against a variable:

$$\pi_j = \begin{cases} \frac{1 + |\{k : W_k \leq -W_j\}|}{p} & \text{if} \quad W_j > 0 \\ 1 & \text{if} \quad W_j \leq 0. \end{cases} \tag{1}$$

In [17] $\pi$ statistics are treated as $p$-values and aggregated using quantile aggregation [15]. However, they can only be considered $p$-values under restrictive assumptions that are hard to check. In the next section, these statistics are used as a building block to reach FDP control. The KOPI framework does not require $\pi$ statistics to be valid $p$-values.

## 4 Main contribution: FDP control for aggregated Knockoffs

### 4.1 Post hoc FDP control for $\pi$ statistics

To obtain FDP control, we rely on Joint Error Rate control as introduced in [4]. For $k_{max} \in [\![p]\!]$, we define a *threshold family* of size $k_{max}$ as a vector $\mathbf{t} = (t_j)_{j \in [\![k_{max}]\!]}$ such that $0 \leq t_1 \leq \cdots \leq t_{k_{max}} \leq 1$.

**Definition 3** (Joint Error Rate, 4)**.** Denote by $\pi_{(j:\mathcal{H}_0)}$ the $j^{th}$ smallest value $\pi_j$ amongst all null hypotheses. The JER associated with $\mathbf{t} = (t_j)_{j \in [\![k_{max}]\!]}$ is:

$$\text{JER}(\mathbf{t}) = \mathbb{P}\left(\exists j \in [\![k_{max} \wedge p_0]\!] : \pi_{(j:\mathcal{H}_0)} < t_j\right). \tag{2}$$

The threshold family $\mathbf{t}$ is said to control the JER at level $\alpha$ iff $\text{JER}(\mathbf{t}) \leq \alpha$.

An $\alpha$-level FDP upper bound can be derived from JER control via the following result:

**Proposition 1** (FDP control via JER control 4)**.** *If $\mathbf{t}$ is a threshold family of length $k_{max}$ that controls the JER at level $\alpha$, then, $V^{\mathbf{t}}(S)/|S|$ is an $\alpha$-level FDP upper bound, with:*

$$V^{\mathbf{t}}(S) = \min_{1 \leq k \leq k_{max}} (k-1) + \sum_{i \in S} 1_{\{\pi_i > t_k\}}. \tag{3}$$

The proof of this result – originally included in [4] – can be found in appendix A.1 for self-containedness. In the remainder of this section, we show how to obtain JER control for $\pi$ statistics.

### 4.2 Joint distribution of $\pi$ statistics under the null

By Definition 3, $\text{JER}(\mathbf{t})$ of a given threshold family only depends on the joint null distribution of the $\pi$ statistics. As for earlier FDR control [1] or k-FWER control [11] results, the key idea to obtain JER control for $\pi$ statistics is to prove that the relevant part of this distribution is in fact known, thanks to the properties of knockoff statistics. We use the same notation as in [11]. Letting $Z_j = |\{k \in [\![p]\!] : W_k \leq -W_j\}|$ and $\chi_j = sign(W_j)$, the $\pi$ statistics $(\pi_j)_{j=[\![p]\!]}$ are given by:

$$\pi_j = \frac{1 + Z_j}{p} 1_{\{\chi_j = 1\}} + 1_{\{\chi_j = -1\}}.$$

For a given $\mathbf{W}$, let $\sigma(\mathbf{W})$ be a permutation of $[\![p]\!]$ that sorts $\mathbf{W}$ by decreasing modulus: $\sigma(\mathbf{W}) = (\sigma_1, \ldots, \sigma_p)$ such that $|W_{\sigma_1}| \geq |W_{\sigma_2}| \cdots \geq |W_{\sigma_p}|$. We start by proving that the $Z$ statistics can be expressed as a function of the vector of $\chi$ statistics:

**Lemma 1.** *For $j \in [\![p]\!]$ such that $\chi_{\sigma_j} = 1$, $Z_{\sigma_j} = \sum_{k=1}^{j-1} 1_{\{\chi_{\sigma_k} = -1\}}$.*

*Proof of Lemma 1.* Since $\chi_{\sigma_j} = 1$, we have:

$$
\begin{aligned}
Z_{\sigma_j} &= |\{k \in [\![p]\!] : W_{\sigma_k} \leq -W_{\sigma_j}\}| \\
&= |\{k \in [\![p]\!] : W_{\sigma_k} < 0 \text{ and } W_{\sigma_k} \leq -W_{\sigma_j}\}| \\
&= |\{k \in [\![p]\!] : W_{\sigma_k} < 0 \text{ and } |W_{\sigma_k}| \leq |W_{\sigma_j}|\}| \\
&= |\{k \in [\![p]\!] : W_{\sigma_k} < 0 \text{ and } k \leq j\}| \\
&= \sum_{k=1}^{j-1} 1_{\{\chi_{\sigma_k} = -1\}}.
\end{aligned}
$$

$\square$

Lemma 1 implies that the distribution of order statistics of $\pi|\sigma(\mathbf{W})$ is entirely determined by that of $\chi|\sigma(\mathbf{W})$. To formalize this, we introduce $\pi^0$ statistics.

**Definition 4** ($\pi^0$ statistics). Let $\chi^0 = (\chi_j^0)_{1 \leq j \leq p}$ be a collection of $p$ i.i.d. Rademacher random variables, that is, for all $j$, $\mathbb{P}(\chi_j^0 = 1) = \mathbb{P}(\chi_j^0 = -1) = 1/2$. The associated $\pi^0$ statistics are defined for $j \in [\![p]\!]$ by

$$
\pi_j^0 = \frac{1 + Z_j^0}{p} 1_{\{\chi_j^0 = 1\}} + 1_{\{\chi_j^0 = -1\}}, \text{ where } Z_j^0 = \sum_{k=1}^{j-1} 1_{\{\chi_k^0 = -1\}}. \tag{4}
$$

**Theorem 1.** *Let $\mathbf{t}$ be a threshold family of length $k_{max}$. Then, for $\pi^0 = (\pi_j^0)_{j \in [\![p]\!]}$ as in (4),*

$$
\text{JER}(\mathbf{t}) \leq \text{JER}^0(\mathbf{t}) := \mathbb{P}\left(\exists k \in [\![k_{max}]\!] : \pi_{(k)}^0 < t_k\right). \tag{5}
$$

*Proof of Theorem 1.* Let $k \in [\![k_{max}]\!]$. Since $t_k \leq 1$, we have $\pi_{(k:\mathcal{H}_0)} < t_k$ if and only if $N_k \geq k$, where

$$
N_k = \left|\left\{j \in \mathcal{H}_0, \chi_j = 1 \text{ and } \frac{1 + Z_j}{p} < t_k\right\}\right|.
$$

With the notation of Definition 4, we define the random variable

$$
N_k^0 = \left|\left\{j \in \mathcal{H}_0, \chi_j^0 = 1 \text{ and } \frac{1 + Z_j^0}{p} < t_k\right\}\right|.
$$

If $\mathcal{H}_0 = [\![p]\!]$, then Lemma 1 implies that conditional on $\sigma(\mathbf{W})$, $N_k$ and $N_k^0$ have the same distribution. Indeed, the vectors $(W_j)_{j/\chi_j=1}$ and $(Z_j)_{j/\chi_j=1}$ have the same ordering, and conditional on $\sigma(\mathbf{W})$, $(\chi_j)_{j \in \mathcal{H}_0}$ are jointly independent and uniformly distributed on $\{-1, 1\}$ (Lemma 2.1 in [11]; [1]). Using the same argument as in the proof of Lemma 3.1 in Janson and Su [11], in the case where $\mathcal{H}_0 \subsetneq [\![p]\!]$, false null $\chi_j$ will insert $-1$'s into the process on the nulls, implying that $N_k$ is stochastically dominated by $N_k^0$. Noting that $N_k^0 \geq k$ if and only if $\pi_{(k)}^0 < t_k$, we obtain that

$$
\mathbb{P}\left(\exists k \in [\![k_{max} \wedge p_0]\!], \pi_{(k:\mathcal{H}_0)} < t_k | \sigma(\mathbf{W})\right) \leq \mathbb{P}\left(\exists k \in [\![k_{max} \wedge p_0]\!], \pi_{(k)}^0 < t_k\right)
$$

$$
\leq \mathbb{P}\left(\exists k \in [\![k_{max}]\!], \pi_{(k)}^0 < t_k\right).
$$

Taking the expectation with respect to $\sigma(\mathbf{W})$ yields the desired result. $\square$

Theorem 1 is related to Lemma 3.1 of Janson and Su [11] and Lemma 3.1 of Li et al. [13], that rely on the sign-flip property of Knockoff statistics under the null [1]. The interest of Theorem 1 is that the upper bound $\text{JER}^0(\mathbf{t})$ only depends on the $\pi^0$ statistics and the threshold family $\mathbf{t}$, and not on the original data. Therefore, it can be estimated with arbitrary precision for any given $\mathbf{t}$ using Monte-Carlo simulation, as explained in the next section and described in Algorithm 1 in Supp. Mat.

### 4.3 Joint Error Rate control for $\pi$ statistics via calibration

To approximate the JER upper bound derived in Theorem 1, we draw $B$ Monte-Carlo samples using Algorithm 1. This yields a set of $B$ vectors of $\pi^0$ statistics denoted by $\pi_b^0 \in \mathbb{R}^p$ for each $b \in [\![B]\!]$. This allows us to evaluate the empirical JER, which estimates the upper bound of interest.

**Definition 5** (Empirical JER). For $B$ vectors of $\pi^0$ statistics and a threshold family $\mathbf{t}$, the empirical JER is defined as:

$$\widehat{\mathrm{JER}}_B^0(\mathbf{t}) = \frac{1}{B} \sum_{b=1}^{B} 1 \left\{ \exists k \in [\![k_{max}]\!] : \pi_{b(k)}^0 < t_k \right\}, \tag{6}$$

where for each $b \in [\![B]\!]$, $\pi_{b(1)}^0 \leq \cdots \leq \pi_{b(p)}^0$.

Since $\widehat{\mathrm{JER}}_B^0(\mathbf{t})$ can be made arbitrarily close (by choosing $B$ large enough) to $\widehat{\mathrm{JER}}^0(\mathbf{t})$ for any given threshold family $\mathbf{t}$, it remains to choose $\mathbf{t}$ such that $\widehat{\mathrm{JER}}^0(\mathbf{t}) \leq \alpha$ in order to ensure JER control. To this end, we consider a sorted set of candidate threshold families called a *template*:

**Definition 6** (Template [4]). A template is a component-wise non-decreasing function $\mathbf{T} : [0, 1] \mapsto \mathbb{R}^p$ that maps a parameter $\lambda \in [0, 1]$ to a threshold family $\mathbf{T}(\lambda) \in \mathbb{R}^p$.

This definition is naturally extended to the case of templates containing a finite number of threshold families. The template corresponding to $B'$ threshold families is then denoted by $(\mathbf{T}(b'/B'))_{b' \in [\![B']\!]}$.

Once a template is specified, the *calibration* procedure [4] can be performed; this consists in finding the least conservative threshold family $\mathbf{t}$ amongst the template that controls the empirical JER at level $\alpha$. Formally, we consider the threshold family defined $\mathbf{t}_\alpha^B = \mathbf{T}(\lambda_B(\alpha))$, where

$$\lambda_B(\alpha) = \frac{1}{B'} \max \left\{ b' \in [\![B']\!] \quad s.t. \quad \widehat{\mathrm{JER}}_B^0 \left( \mathbf{T}\left(\frac{b'}{B'}\right) \right) \leq \alpha \right\}.$$

As observed by Blain et al. [3], optimal power is reached when the candidate families match the shape of the distribution of the null statistics. We define a template based on the distribution of the $\pi^0$ statistics appearing in Theorem 1. In practice, we draw $B'$ samples from this distribution independently from the $B$ Monte Carlo samples to avoid circularity biases. Since a template has to be component-wise non-decreasing, i.e. the set of candidate threshold families has to be sorted, we extract empirical quantiles from these $B'$ sorted vectors. This yields a template $\mathbf{T}^0$ composed of $B'$ candidate curves that match quantiles of the distribution of $\pi^0$ statistics. The $\frac{b'}{B'}$-quantile curve defines the threshold family $\mathbf{T}^0(b'/B')$. We obtain the following result:

**Theorem 2** (JER control for $\pi$-statistics). *Consider the threshold family defined by $\mathbf{t}_\alpha^B = \mathbf{T}^0(\lambda_B(\alpha))$. Then, as $B \to +\infty$,*

$$\mathrm{JER}(\mathbf{t}_\alpha^B) \leq \alpha + O_P(1/\sqrt{B}).$$

The number $B$ of Monte-Carlo samples in Theorem 2 can be chosen arbitrarily large to obtain JER control, leading to valid FDP bounds via Equation 3. This result is proved in Appendix A.2.

### 4.4 False Discovery Proportion control for aggregated Knockoffs

In the previous section we have seen how to reach FDP control via Knockoffs. As explained above, aggregation is needed to mitigate the randomness of the Knockoff generation process. Therefore, we aim to extend the previous result to the case of aggregated Knockoffs. Let us first define aggregation:

**Definition 7.** For $D$ draws of Knockoffs, an aggregation procedure is a function $f : \mathbb{R}^D \mapsto \mathbb{R}$ that maps a vector of $(\pi^d)_{d \in [\![D]\!]}$ statistics to an aggregated statistic $\overline{\pi}$.

In practice, since we have $p$ variables, aggregation is performed for each variable, i.e.:

$$\forall j \in [\![p]\!], \quad f(\pi_j^1, \ldots, \pi_j^D) = \overline{\pi_j}.$$

Then, inference is performed on the vector of aggregated statistics $(\overline{\pi}_1, \ldots, \overline{\pi}_p)$.

For a fixed aggregation scheme $f$, we can naturally extend the calibration procedure of the preceding section. Instead of drawing a single $B \times p$ matrix of $\pi^0$ statistics containing $\pi_b^0 \in \mathbb{R}^p$ for each $b \in [\![B]\!]$, we draw $D$ such matrices. Given $d \in [\![D]\!]$, each matrix contains $\pi_b^{0,d} \in \mathbb{R}^p$ for each $[\![B]\!]$.

Then, for each $b \in [\![B]\!]$, we perform aggregation: $\overline{\pi}_b^0 = f\left( (\pi_b^{0,d})_{d \in [\![D]\!]} \right)$. The JER in the aggregated case is defined as:
$$\overline{\mathrm{JER}}(\mathbf{t}) = \mathbb{P}\left( \exists j \in [\![k_{max} \wedge p_0]\!] : \overline{\pi}_{(j:\mathcal{H}_0)} < t_j \right).$$

We obtain the aggregated template following the same procedure, i.e. drawing $D$ templates and aggregating them. For each $b' \in [\![B']\!]$, the aggregated threshold family is written:
$$\overline{\mathbf{T}}\left( \frac{b'}{B'} \right) = f\left( \left( \mathbf{T}^d \left( \frac{b'}{B'} \right) \right)_{d \in [\![D]\!]} \right).$$

We can then write the empirical JER in the aggregated case as:
$$\widehat{\overline{\mathrm{JER}}}\left( \overline{\mathbf{T}}\left( \frac{b'}{B'} \right) \right) = \frac{1}{B} \sum_{b=1}^{B} \mathbb{1}\left\{ \exists j \in [\![k_{max}]\!] : \overline{\pi}_{b(j)}^0 < \overline{\mathbf{T}}_j \left( \frac{b'}{B'} \right) \right\}.$$

Calibration can be performed in the same way as in the non-aggregated case. Note that we perform calibration *after* aggregating; therefore, JER control is ensured directly on aggregated statistics and is not a result of aggregating JER controlling families. Importantly, this approach holds without additional assumptions on the aggregation scheme $f$. We consider the threshold family $\overline{\mathbf{t}}_\alpha^B = \overline{\mathbf{T}}(\lambda_B(\alpha))$, where
$$\lambda_B(\alpha) = \frac{1}{B'} \max\left\{ b' \in [\![B']\!] \quad s.t. \quad \widehat{\overline{\mathrm{JER}}}_B \left( \overline{\mathbf{T}}\left( \frac{b'}{B'} \right) \right) \leq \alpha \right\}.$$

With $\overline{\mathbf{T}}^0$ a template composed of $B'$ candidate curves that match quantiles of the distribution of $\overline{\pi}^0$ statistics, we obtain the following result:

**Theorem 3** (JER control for aggregated $\pi$-statistics)**.** *Consider the threshold family defined by* $\overline{\mathbf{t}}_\alpha^B = \overline{\mathbf{T}}^0(\lambda_B(\alpha))$. *Then, as* $B \to +\infty$,
$$\overline{\mathrm{JER}}(\overline{\mathbf{t}}_\alpha^B) \leq \alpha + O_P(1/\sqrt{B}).$$

*Proof.* The proof is identical to that of Theorem 2 using the empirical aggregated JER. $\qquad\square$

The calibrated aggregated threshold family yields valid FDP upper bounds via Proposition 1. The proposed **KOPI** (Knockoffs - $\pi$) method therefore achieves FDP control on aggregated Knockoffs.

## 5 Experiments

**Methods considered.** In our implementation of KOPI, we rely on the harmonic mean [28] as the aggregation scheme $f$. Additionally, we set $k_{max} = \lfloor p/50 \rfloor$ following the approach of [3]. We also consider both state-of-the-art Knockoffs aggregation schemes: AKO (Aggregation of Multiple Knockoffs, 17) and e-values based aggregation [18]. Additionally, we consider Vanilla Knockoffs, i.e. [6] and FDP control via Closed Testing [13]. In simulated data experiments, we generate Knockoffs assuming a Gaussian distribution for $\mathbf{X}$, with all variables centered. For methods that support aggregation, we use $D = 50$ Knockoff draws.

### 5.1 Simulated data

**Setup.** At each simulation run, we generate Gaussian data $\mathbf{X} \in \mathbb{R}^{n \times p}$ with a Toeplitz correlation matrix corresponding to a first-order auto-regressive model with parameter $\rho$, i.e. $\mathbf{\Sigma}_{i,j} = \rho^{|i-j|}$.

Then, we draw the true support $\boldsymbol{\beta}^* \in \{0,1\}^p$. The number of non-null coefficients of $\boldsymbol{\beta}^*$ is controlled by the sparsity parameter $s_p$, i.e. $s_p = \|\boldsymbol{\beta}^*\|_0/p$. The target variable $\mathbf{y}$ is built using a linear model:
$$\mathbf{y} = \mathbf{X}\boldsymbol{\beta}^* + \sigma\boldsymbol{\epsilon},$$

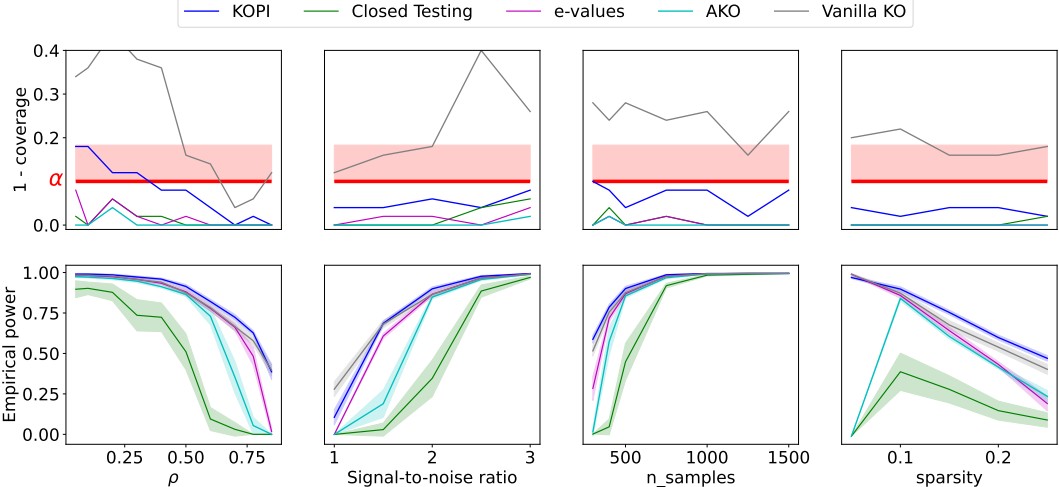

Figure 1: **FDP bound coverage at level $\alpha$ and empirical Power for 50 simulation runs and five different methods:** Vanilla Knockoffs, aggregated Knockoffs using e-values, aggregated Knockoffs using quantile-aggregation, KOPI and Knockoff inference via Closed Testing. We use $D = 50$ Knockoffs draws and the following simulation settings: $\alpha = 0.1, q = 0.1, p = 500$. Each column represents a varying parameter with the first row displaying FDP coverage and the second row displaying power. The red line and associated error bands represent the acceptable limits for FDP bound coverage. KOPI consistently outperforms all other methods while retaining FDP control.

with $\sigma$ controlling the amplitude of the noise: $\sigma = \|\mathbf{X}\boldsymbol{\beta}^*\|_2/(\text{SNR}\|\boldsymbol{\epsilon}\|_2)$, SNR being the signal-to-noise ratio. We choose the central setting $n = 500, p = 500, \rho = 0.5, s_p = 0.1, \text{SNR} = 2$. For each parameter, we explore a range of possible values to benchmark the methods across varied settings.

To select variables using FDP upper bounds, we retain the largest possible set of variables $S$ such that $V(S) \leq q|S|$ (Algorithm 4). For each of the $N$ simulations and each method, we compute the empirical FDP and True Positive Proportion (TPP):

$$\widehat{FDP}(S) = \frac{|S \cap \mathcal{H}_0|}{|S|} \quad \text{and} \quad \widehat{TPP}(S) = \frac{|S \cap \mathcal{H}_1|}{|\mathcal{H}_1|}.$$

If the FDP is controlled at level $\alpha$, $|\{k \in [\![N]\!] : \widehat{FDP}(S_k) > q\}| \sim \mathcal{B}(N, \alpha)$. Then, we can compute error bands on the $\alpha$-level using $\text{std}\left(\mathcal{B}(N, \alpha)/N\right) = \sqrt{\alpha(1 - \alpha)/N}$. The second row of Fig. 1 represents the empirical power achieved by each method, which corresponds to the average of TPPs defined above for $N$ runs i.e. Power $= \sum_{k=1}^{N} \widehat{TPP}(S_k)/N$. Fig. 1 shows that across all different settings, KOPI retains FDP control. We can also see that FDR control does not imply FDP control, as Vanilla Knockoffs are consistently outside of FDP bound coverage intervals. However, the two existing aggregation schemes (AKO and e-values) that formally guarantee FDR control are generally conservative and achieve FDP control empirically. This is consistent with the findings of [18]. The Closed Testing procedure of [13] achieves FDP control as announced but suffers from a lack of power.

Interestingly, KOPI achieves FDP control while offering power gains compared to FDR-controlling Knockoffs aggregation methods. Yet FDP control is a much stronger guarantee than FDR control, as discussed previously. These gains are especially noticeable in challenging inference settings where most methods exhibit a clear decrease in power or even catastrophic behavior (i.e. zero power).

Moreover, Fig. 3 (in appendix) shows that when using $q = 0.05$ rather than $q = 0.1$ as in Fig. 1, the robustness of KOPI with regards to difficult inference settings is even more salient. More precisely, for $q = 0.05$, AKO and Closed Testing are always powerless. E-values aggregation yields good power in easier settings such as $\rho \leq 0.6$, SNR $\geq 2.5$ or $n > 750$ but exhibits catastrophic behavior in harder settings. Overall, apart from KOPI, only Vanilla Knockoffs exhibit non-zero power, but this method fails to control the FDP as it is intended to control FDR. KOPI preserves FDP control in all settings while yielding superior power compared to all other methods.

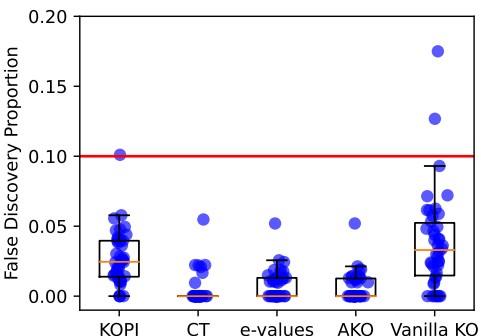 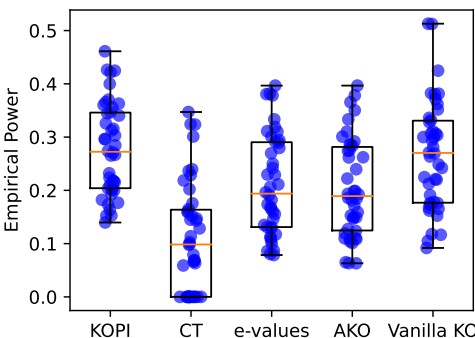

Figure 2: **Empirical FDP and power on semi-simulated data for 42 contrast pairs.** We use 7 HCP contrasts C0: "Motor Hand", C1: "Motor Foot", C2: "Gambling", C3: "Relational", C4: "Emotion", C5: "Social", C6: "Working Memory". We consider all 42 possible train/test pairs: the train contrast is used to obtain a ground truth, while the test contrast is used to generate the response. Inference is performed using the 5 methods considered in the paper and the empirical FDP is reported in the left box plot, while power is reported in the right box plot. Notice (right figure) that KOPI yields superior power compared to all other Knockoffs-based methods while controlling the FDP (left Fig.).

## 5.2 Brain data application

The goal of human brain mapping is to associate cognitive tasks with relevant brain regions. This problem is tackled using functional Magnetic Resonance Imaging (fMRI), which consists in recording the blood oxygenation level dependent signal via an MRI scanner. The importance of conditional inference for this problem has been outlined in [26]. We use the Human Connectome Project (HCP900) dataset that contains brain images of healthy young adults performing different tasks while inside an MRI scanner. Details about this dataset and empirical results can be found in Appendix E.

While these results demonstrate the face validity of the approach, FDP control and power cannot be evaluated. Therefore, following [16], we consider an additional experiment that consists in using semi-simulated data. We consider a first fMRI dataset $(\mathbf{X}_1, \mathbf{y}_1)$ on which we perform inference using a Lasso estimator; this yields $\boldsymbol{\beta}_1^* \in \mathbb{R}^p$ that we will use as our ground truth. Then, we consider a separate fMRI dataset $(\mathbf{X}_2, \mathbf{y}_2)$ for data generation. The point of using a separate dataset is to avoid circularity between the ground truth definition and the inference procedure. Concretely, we discard the original response vector $\mathbf{y}_2$ for this dataset and build a simulated response $\mathbf{y}_2^{sim}$ using a linear model, with the same notation as previously (we set $\sigma$ so that $SNR = 4$): $\mathbf{y}_2^{sim} = \mathbf{X}_2 \boldsymbol{\beta}_1^* + \sigma \boldsymbol{\epsilon}$.

Then, inference is performed using Knockoffs-based methods on $(\mathbf{X}_2, \mathbf{y}_2^{sim})$. Since we consider $\boldsymbol{\beta}_1^*$ as the ground truth, the FDP and TPP can be computed for each method. As can be seen in Fig. 2, KOPI is the most powerful method among those that control the FDP.

## 5.3 Genomic data application

In addition to the brain data application, we compared KOPI to other Knockoffs-based methods on gene-expression data [5] containing 79 samples and 90 genes. KOPI yields a non trivial selection for all runs, with 3 genes selected in $100\%$ of all 50 runs of the experiment. Across all runs, only 8 different genes are selected by KOPI. Vanilla Knockoffs select 24 different genes across all runs and no gene exceeds a selection frequency of $70\%$. All other methods are powerless in all runs. Details and results of this experiment can be found in Appendix D.

## 6 Discussion

In this paper, we have proposed a novel method that reaches FDP control on aggregated Knockoffs. It combines the benefits of aggregation, i.e. improving the stability of the inference, in addition to providing a probabilistic control of the FDP, rather than controlling only its expectation, the FDR.

Simulation results support that KOPI indeed controls the FDP. Furthermore, while FDP control is a stricter guarantee than FDR control, KOPI actually offers power gains compared to state-of-the-art aggregation-based Knockoffs methods. This sensitivity gain is a direct benefit from the JER approach and its adaptivity to arbitrary aggregation schemes. While the latter has been formulated and used so far in mass univariate settings [3], the present work presents a first use of this approach in the context of multiple regression. Moreover, KOPI does not require any assumption on the data at hand or on the law of Knockoff statistics under the null.

The computation time of the proposed approach is comparable to existing aggregation schemes for Knockoffs: sampling $\pi$ statistics under the null using Algorithm 1 can be done once and for all for a given value of $p$. JER estimation via Algorithm 2 and calibration can be performed via binary search of complexity $\mathcal{O}(log(B'))$. Finding the rejection set $\hat{S}$ after performing calibration is done in linear time via [7]. In practice, the computation time is the same as for classical knockoff aggregation [19] and is in minutes for the brain imaging datasets considered. Avenues for future work include a theoretical analysis of the False Negative Proportion (FNP) [8] of KOPI and developing a step-down version of the method to further improve power.

We provide a Python package containing the code for KOPI available at `https://github.com/alexblnn/KOPI`.

# 7 Acknowledgments and disclosure of funding

This project was funded by a UDOPIA PhD grant from Université Paris-Saclay and also supported by the FastBig ANR project (ANR-17-CE23-0011), the KARAIB AI chair (ANR-20-CHIA-0025-01), the H2020 Research Infrastructures Grant EBRAIN-Health 101058516 and the SansSouci ANR project (ANR-16-CE40-0019). The authors thank Binh Nguyen for his precious help on the code base and Samuel Davenport for useful discussions about this work.

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
