# Appendices

# A Proofs

## A.1 Proof of FDP control via JER control

For self-containedness we provide a Proof of Proposition 1 adapted from [4]:

**Proposition 1** (FDP control via JER control 4)**.** *If* $\mathbf{t}$ *is a threshold family of length* $k_{max}$ *that controls the* JER *at level* $\alpha$*, then,* $V^{\mathbf{t}}(S)/|S|$ *is an* $\alpha$*-level FDP upper bound, with:*

$$V^{\mathbf{t}}(S) = \min_{1 \leq k \leq k_{max}} (k-1) + \sum_{i \in S} 1_{\{\pi_i > t_k\}}. \tag{3}$$

*Proof.* Denote by $R_k = \{j : \pi_j \leq t_k\}$. Then for any set $S$:

$$
\begin{aligned}
|S \cap H_0| &= \left| S \cap \overline{R_k} \cap H_0 \right| + |S \cap R_k \cap H_0| \\
&\leq \left| S \cap \overline{R_k} \right| + |R_k \cap H_0| \\
&= \sum_{i \in S} 1_{\{\pi_i(X) > t_k\}} + |R_k \cap H_0| \\
&\leq \sum_{i \in S} 1_{\{\pi_i(X) > t_k\}} + k - 1 \\
&=: V_k^{\mathbf{t}}(S),
\end{aligned}
$$

where the last inequality holds with probability at least $1 - \alpha$ by (2). Since (2) holds simultaneously for all $k$, the minimum over $k$ of all $V_k(S)$ is an $\alpha$-level upper bound on the false positives in $S$ and therefore $V^{\mathbf{t}}(S)/|S|$ is itself an $\alpha$-level FDP upper bound. $\square$

## A.2 Proof of Theorem 2

**Lemma 2.** *For any threshold family* $\mathbf{t}$*, we have*

$$\mathrm{JER}^0(\mathbf{t}) - \widehat{\mathrm{JER}}_B^0(\mathbf{t}) = O_P(1/\sqrt{B})$$

*Proof of Lemma 2.* Let $Z_B(\mathbf{t}) = \sqrt{B}\left(\mathrm{JER}^0(\mathbf{t}) - \widehat{\mathrm{JER}}_B^0(\mathbf{t})\right)$. By the Central Limit Theorem, we have

$$Z_B(\mathbf{t}) \xrightarrow[B \to \infty]{d} Z(\mathbf{t}),$$

where $Z(\mathbf{t})$ is a centered Gaussian random variable with variance $\sigma^2(\mathbf{t}) = \mathrm{JER}^0(\mathbf{t})(1 - \mathrm{JER}^0(\mathbf{t}))$. As such, for any $M > 0$, we have

$$\mathbb{P}\left(|Z_B(\mathbf{t})| \geq M\right) \xrightarrow[B \to \infty]{} \mathbb{P}\left(|Z(\mathbf{t})| \geq M\right).$$

Since $\mathrm{JER}^0(\mathbf{t}) \leq 1$, we have $\sigma^2(\mathbf{t}) \leq 1/4$ for any $\mathbf{t}$, so that $Z(\mathbf{t})$ is stochastically dominated by $\mathcal{N}(0, 1/4)$, which does not depend on the threshold family $\mathbf{t}$. As such, we have $\mathbb{P}\left(|Z(\mathbf{t})| \geq M\right) = 2\mathbb{P}\left(Z(\mathbf{t}) \geq M\right) \leq 2\overline{\Phi}(2M)$, where $\overline{\Phi}$ denotes the tail function of the standard normal distribution. Since $\overline{\Phi}(x)$ tends to 0 as $x \to +\infty$, we have proved that $Z_B(\mathbf{t}) = O_P(1)$. $\square$

**Theorem 2** (JER control for $\pi$-statistics)**.** *Consider the threshold family defined by* $\mathbf{t}_\alpha^B = \mathbf{T}^0(\lambda_B(\alpha))$*. Then, as* $B \to +\infty$*,*

$$\mathrm{JER}(\mathbf{t}_\alpha^B) \leq \alpha + O_P(1/\sqrt{B}).$$

*Proof.* We treat the case where $\mathbf{t}_\alpha^B$ is well defined for all $B$, i.e. that there exists a threshold family amongst $\mathbf{T}^0$ controls the empirical $\mathrm{JER}^0$ for $B$ draws. If this is not the case for some $B$, then $\mathbf{t}_\alpha^B$ is set to the null family and the result holds.

By Theorem 1 we have for all $\mathbf{t}$ that $\mathrm{JER}\left(\mathbf{t}\right) \leq \mathrm{JER}^0\left(\mathbf{t}\right)$. We can write:

$$\mathrm{JER}^0\left(\mathbf{t}\right) = \widehat{\mathrm{JER}}_B^0(\mathbf{t}) + \left(\mathrm{JER}^0\left(\mathbf{t}\right) - \widehat{\mathrm{JER}}_B^0(\mathbf{t})\right)$$

$$= \widehat{\mathrm{JER}}_B^0(\mathbf{t}) + O_P(1/\sqrt{B})$$

by Lemma 2. Applying the above to $\mathbf{t} = \mathbf{t}_\alpha^B$ yields the desired result since $\widehat{\mathrm{JER}}_B^0(\mathbf{t}_\alpha^B) \leq \alpha$ by definition. $\qquad\square$

## B  Algorithms

Algorithm 1 describes the procedure to obtain samples from the joint distribution $(\pi_k^0)_k$. This is useful to compute the empirical JER of Equation 6 via 2 and in turn to perform calibration which is described in Algorithm 3. Once calibration is performed, inference can be performed using Algorithm 4. The FDP of resulting regions is provably controlled thanks to Theorem 3.

---

**Algorithm 1: Sampling from the joint distribution of $\pi$ statistics under the null** according to Theorem 1.

---

1  **Input:** $B$ the number of MC draws; $p$ the number of variables
2  **Output:** $\mathbf{\Pi}_0 \in [0,1]^{B \times p}$ a matrix of $\pi^0$ statistics
3  $\mathbf{\Pi}_0 \leftarrow \mathrm{zeros}(B, p)$
4  **for** $b \in [1, B]$ **do**
5  $\quad$ $\chi \leftarrow \mathrm{draw\_random\_vector}(\{-1, 1\}^p)$ // Draw signs
6  $\quad$ $Z = 0$ // Initialize count
7  $\quad$ **for** $j \in [1, p]$ **do**
8  $\quad\quad$ **if** $\chi[j] < 0$ **then**
9  $\quad\quad\quad$ $\mathbf{\Pi}_0[b][j] \leftarrow 1$
10 $\quad\quad\quad$ $Z \leftarrow Z + 1$ // Increment $Z$
11 $\quad\quad$ **end**
12 $\quad\quad$ **else**
13 $\quad\quad\quad$ $\mathbf{\Pi}_0[b][j] \leftarrow \frac{1+Z}{p}$
14 $\quad\quad$ **end**
15 $\quad$ **end**
16 **end**
17 $\mathbf{\Pi}_0 \leftarrow \mathrm{sort\_lines}(\mathbf{\Pi}_0)$ // Sort samples
18 **Return** $\mathbf{\Pi}_0$

---

**Algorithm 2: Computing the Empirical JER.** The empirical JER is computed for a given threshold family and a matrix of $\pi^0$ statistics. This algorithm is similar to Algorithm 3 of [3].

---

1  **Input:** $\mathbf{\Pi}_0$ a matrix of $\pi^0$ statistics; $\mathbf{t}$ a threshold family; $k_{max}$ the size of the threshold family
   **Output:** $\widehat{\mathrm{JER}}$, the empirical JER of threshold family $\mathbf{t}$
2  $(B, \mathrm{p}) \leftarrow \mathrm{shape}(\mathbf{\Pi}_0)$
3  $\widehat{\mathrm{JER}} \leftarrow 0$
4  **for** $b \in [1, B]$ **do**
5  $\quad$ **for** $i \in [1, k_{max}]$ **do**
6  $\quad\quad$ $\mathrm{diff}[i] \leftarrow \mathbf{\Pi}_0[b'][i] - \mathbf{t}[i]$
7  $\quad\quad$ // Check JER control at rank $i$
8  $\quad$ **end**
9  $\quad$ **if** $\min(\mathrm{diff}) < 0$ **then**
10 $\quad\quad$ $\widehat{\mathrm{JER}} \leftarrow \widehat{\mathrm{JER}} + \frac{1}{B}$
11 $\quad\quad$ // Increment risk if JER control event is violated
12 $\quad$ **end**
13 **end**
14 **Return** $\widehat{\mathrm{JER}}$

---

**Algorithm 3: Performing calibration on $\pi$-statistics**. First, we use Theorem 1 to build a suitable template and estimate the JER of each candidate threshold family. Then, we perform calibration to select the least conservative possible threshold family that controls the JER at a given level $\alpha$.

---

1 **Input:** $\alpha$ the desired FDP coverage; $B$ the number of MC draws for JER estimation; $B'$ the number of candidate threshold families
2 **Output:** $\mathbf{t}_\alpha$ the calibrated threshold family at level $\alpha$
3 $\mathbf{\Pi}_0 \leftarrow$ draw_null_$\pi(B, p)$ // `Algorithm 1`
4 $\mathbf{\Pi}_0' \leftarrow$ draw_null_$\pi(B', p)$
5 **for** $b' \in [1, B']$ **do**
6      $\mathbf{T}[b'] \leftarrow$ quantiles$(\mathbf{\Pi}_0', \frac{b'}{B'})$ // `Build template`
7      $\widehat{\mathrm{JER}}_{b'} \leftarrow$ empirical_jer$(\mathbf{\Pi}_0, \mathbf{T}[b'])$ // `Apply Algorithm 2 for each family`
8 **end**
9 $b'_{cal} \leftarrow \max\{b' \in [1, B'] \text{ s.t. } \widehat{\mathrm{JER}}_{b'} \leq \alpha\}$ // `Perform calibration`
10 $\mathbf{t}_\alpha \leftarrow \mathbf{T}[b'_{cal}]$
11 **Return** $\mathbf{t}_\alpha$

---

**Algorithm 4: Performing inference via Knockoffs and calibration.** We compute the largest possible region that satisfies the required FDP level $q$ using the JER controlling family computed via Algorithm 3. The bound $V^{\mathbf{t}_\alpha}$ is computed from $\pi$ using Equation 3.

---

1 **Input:** $\mathbf{X}$ the input data; $\mathbf{y}$ the target variable; $q$ the maximum tolerable FDP; $\mathbf{t}_\alpha$ the calibrated threshold family at level $\alpha$
2 **Output:** $\hat{S}$ the selected variables
3 $n, p \leftarrow$ shape$(\mathbf{X})$ // `n samples, p variables`
4 $\tilde{\mathbf{X}} \leftarrow$ sample_Knockoffs$(\mathbf{X})$
5 $\mathbf{W} \leftarrow LCD(\mathbf{X}, \tilde{\mathbf{X}}, \mathbf{y})$ // `Compute W`
6 $\pi \leftarrow$ compute_proportion$(\mathbf{W})$ // `Equation (1)`
7 $\hat{S} \leftarrow \max_S \{|S| \quad s.t. \quad \dfrac{V^{\mathbf{t}_\alpha}(S)}{|S|} \leq q\}$ // `Find largest admissible region`
8 // $V^{\mathbf{t}_\alpha}(S)$ `depends on` $\pi$
9 **Return** $\hat{S}$

---

## C   Additional simulation results

### C.1   A harder inference setup

We evaluated the performance of all five methods in the more challenging setting $q = 0.05$ instead of using $q = 0.1$. The results are presented in Fig. 3. In this setting, AKO and Closed Testing are always powerless and aggregation via e-values suffers from a lack of power in most cases. Vanilla Knockoffs exhibit satisfactory power but consistently fail to control the FDP. KOPI preserves FDP control and yields acceptable power.

### C.2   Impact of aggregation scheme choice

While the theoretical guarantees we obtain hold for all choices of aggregation schemes, these hyperparameter impacts the power of KOPI. To assess this, we use the same simulated data setup as in Figure 1 to compare four aggregation schemes: arithmetic mean, geometric mean, harmonic mean and quantile aggregation.

Importantly, we first check that the FDP is controlled for all types of aggregation and in all settings considered by reporting the bound non-coverage. We use three settings of varying difficulty, parametrized by the correlation level $\rho$ and use $\alpha = 0.1, q = 0.1$:

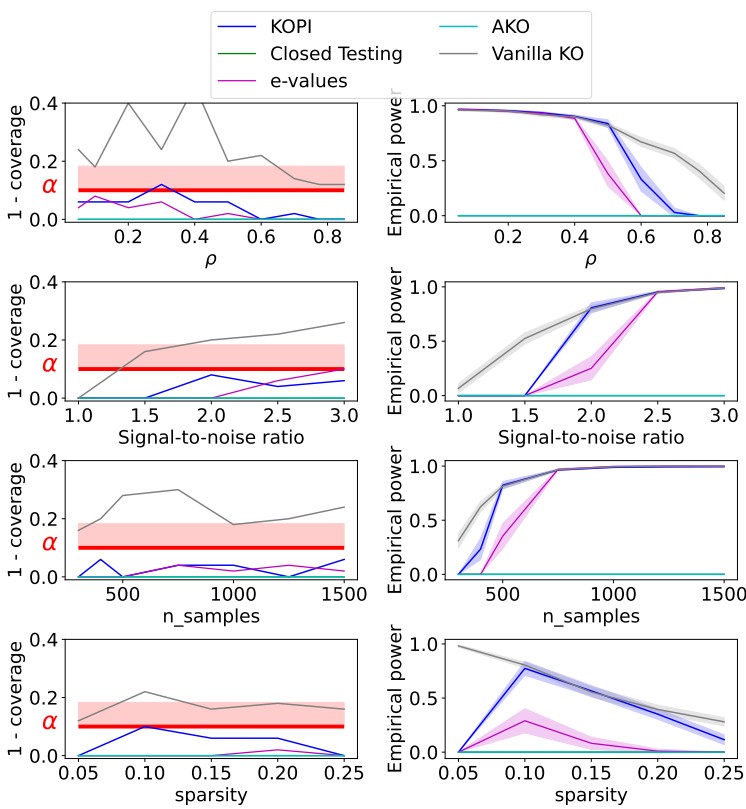

Figure 3: **FDP bound coverage at level $\alpha$ and empirical Power for 50 simulation runs and five different methods.** The five methods are Vanilla Knockoffs, aggregated Knockoffs using e-values, aggregated Knockoffs using quantile-aggregation, KOPI and Knockoff inference via Closed Testing. We use 50 Knockoffs draws and the following simulation setting $\alpha = 0.1, q = 0.05, p = 500$. Each row represents a varying parameter with the left panel displaying FDP coverage and the right panel displaying power. The red line and associated error bands represent the acceptable limits for FDP bound coverage. Notice that KOPI consistently outperforms all other methods while retaining FDP control.

|  | Harmonic | Arithmetic | Geometric | Quantile aggregation |
|---|---|---|---|---|
| $\rho = 0.5$ | 10% | 0% | 2% | 10% |
| $\rho = 0.6$ | 2% | 0% | 0% | 4% |
| $\rho = 0.7$ | 2% | 0% | 0% | 0% |

Table 1: **FDP control of KOPI for four aggregation schemes and three different correlation levels.** Note that FDP control is maintained in all scenarios which is coherent with the result obtained in Theorem 3.

The FDP is indeed controlled in all cases since non-coverage never exceeds the chosen level $\alpha = 10\%$ as seen in Table 1. This is coherent with the theoretical guarantees we obtain in Theorem 3. We now report the average power to benchmark aggregation schemes:

|  | Harmonic | Arithmetic | Geometric | Quantile aggregation |
|---|---|---|---|---|
| $\rho = 0.5$ | **0.91** | 0.77 | 0.87 | 0.90 |
| $\rho = 0.6$ | **0.83** | 0.58 | 0.77 | **0.83** |
| $\rho = 0.7$ | **0.72** | 0.39 | 0.61 | **0.72** |

Table 2: **Empirical power of KOPI for four aggregation schemes and three different correlation levels.** Note that harmonic mean aggregation consistently outperforms arithmetic aggregation and geometric aggregation. Quantile aggregation performs similarly to harmonic aggregation.

Note that harmonic mean aggregation outperforms arithmetic and geometric mean consistently and performs similarly to quantile aggregation as seen in Table 2.

# D   Details and results on genomic data

## D.1   Lymphomatic leukemia mutation classification

Differential gene expression studies aim at identifying genes whose activity differs significantly between two (or more) populations, based on a sample of measurements from individuals from these populations. The activity of a gene is usually quantified by its level of expression in the cell. We consider a microarray data set studied in [5] that consists of expression measurements for biological samples from $n = 79$ individuals with B-cell acute lymphoblastic leukemia (ALL): 37 of these individuals harbor a specific mutation called BCR/ABL, while the remaining 42 do not. Our goal here is to identify, from this sample, genes for which there is a difference in the mean expression level between the mutated and non-mutated populations. We focus on the $p = 90$ genes on chromosome 7 whose individual standard deviation is above $0.5$.

The genes selected by different Knockoffs-based methods are summarized in Table 3. Stability selection criteria analogous to [14, 19] are displayed. Note that the selection made by KOPI is more robust than that of Vanilla Knockoffs: 4 genes are selected in nearly all runs by KOPI, while none are selected as frequently by Vanilla Knockoffs. Conversely, KOPI only selects 2 genes in less than $50\%$ of all runs compared to 18 for Vanilla Knockoffs. This confirms that error control guarantees of KOPI, together with the stability brought by aggregation, lead to avoiding most spurious/non-reproducible detections. Besides KOPI and Vanilla Knockoffs, all other methods are powerless in all runs.

|  | KOPI | Vanilla KO | e-values | Closed Testing | AKO |
|---|---|---|---|---|---|
| Selected in >90% of runs | 4 | 0 | 0 | 0 | 0 |
| Selected in >50% of runs | 6 | 6 | 0 | 0 | 0 |
| Spurious detections (<50% of runs) | 2 | 18 | 0 | 0 | 0 |

Table 3: **Stability selection criteria for 5 Knockoffs-based methods on "Lymphomatic leukemia mutation" genomic data.** Note that KOPI displays a very stable selection set across all runs with 4 genes present in $> 90\%$ of runs. KOPI also avoids most spurious discoveries, as only 2 genes are selected less than $50\%$ of the time, compared to 18 genes using Vanilla Knockoffs. The 6 genes selected more than $50\%$ of the time by KOPI and Vanilla Knockoffs are the same. All other Knockoffs-based methods are powerless in all runs.

## D.2   Colon vs Kidney classification

We also considered an additional genomic dataset to reproduce these results with a larger number of samples. The dataset we used is part of **GEMLeR (Gene Expression Machine Learning Repository)** [20], a collection of gene expression datasets that can be used to benchmark ML methods on genomics data.

We chose the "Colon vs Kidney" dataset: this is a binary classification dataset where the goal is to distinguish cancerous tissue from two different organs (Colon and Kidney) using gene expression data. This dataset comprises 546 samples and 10936 genes. To make the problem tractable for Knockoffs-based methods we perform dimensionality reduction to select the 546 genes that have the largest variance. Then, **we run all Knockoffs-based methods 50 times** and report the selected genes.

The genes selected by different Knockoffs-based methods are summarized in Table 4. Stability selection criteria analogous to [14, 19] are displayed. Note that the selection made by KOPI is more robust than that of Vanilla Knockoffs: 21 genes are selected in nearly all runs by KOPI, while none are selected as frequently by Vanilla Knockoffs. Conversely, KOPI only selects 7 genes in less than $50\%$ of all runs compared to 34 for Vanilla Knockoffs and 20 for e-values aggregation.

| | KOPI | Vanilla KO | e-values | Closed Testing | AKO |
|---|---|---|---|---|---|
| Selected in >90% of runs | 21 | 0 | 0 | 0 | 0 |
| Selected in >50% of runs | 22 | 25 | 0 | 0 | 0 |
| Spurious detections (<50% of runs) | 7 | 34 | 20 | 0 | 0 |

Table 4: **Stability selection criteria for 5 Knockoffs-based methods on "Colon vs Kidney" genomic data.** Note that KOPI displays a very stable selection set across all runs with 21 genes present in $> 90\%$ of runs. KOPI also avoids most spurious discoveries, as only 7 genes are selected less than $50\%$ of the time, compared to 34 genes using Vanilla Knockoffs and 20 using e-values. All other Knockoffs-based methods are powerless in all runs.

# E    Details and results on HCP data

## E.1    HCP dataset

We use the HCP900 task-evoked fMRI dataset [23], in which we take the masked $2\,\mathrm{mm}$ resolution z-statistics maps of the 778 subjects from 7 tasks to solve binary regression problems, namely predicting which condition is associated with the brain image: emotion (*emotional face* vs *shape outline*), gambling (*reward* vs *loss*), language (*story* vs *math*), motor hand (*left* vs *right* hand), motor foot (*left* vs *right* foot), relational (*relational* vs *match*) and social (*mental interaction* vs *random interaction*).

We consider the fixed-effect maps (average across right-left and left-right phase encoding schemes) for each condition, yielding one image per subject per condition (which corresponds to two images per subject for each classification problem). Then, for each problem, the number of samples available is 1556 ($= 2 \times 778$) and the number of voxels is 156 374 after gray-matter masking. Dimension reduction was carried out using Ward parcellation scheme to $1k$ clusters, which is known to yield spatially homogeneous regions [21]. The signal is then averaged per cluster, yielding a reduced design matrix $\mathbf{X}$ for the problem.

## E.2    Brain data are non-Gaussian

In the synthetic data experiments we used the Gaussian Knockoff generation process described in 6. However, fMRI brain maps can be heavily non-Gaussian. In turn, Gaussian Knockoffs cannot satisfy the Knockoffs exchangeability assumption and any statistical control on False Discoveries is rendered spurious.

To build non-Gaussian Knockoffs, we use a linear variant of the Sequential Conditional Independent Pairs (SCIP) algorithm of 6:

---
**Algorithm 5:** Generating Non-Gaussian Knockoffs using the Sequential Conditional Independent Pairs algorithm of 6.

---
1 **for** $j \in [1, p]$ **do**
2 $\quad$ Fit a Lasso model on $(\mathbf{X_{-j}}, X_j)$
3 $\quad$ Compute the residual $\epsilon_j = X_j - \mathbf{X_{-j}}\widehat{\boldsymbol{\beta}}_j$
4 **end**
5 **for** $j \in [1, p]$ **do**
6 $\quad$ Sample $\tilde{X}_j$ from $\mathbf{X_{-j}}\widehat{\boldsymbol{\beta}}_j + \epsilon_{\rho(j)}$ `//` $\rho$ `is a random ordering of` $[1, p]$
7 **end**
8 **Return** $\tilde{\mathbf{X}}_{1:\mathbf{p}}$

---

## E.3    Additional results

The results corresponding to 7 contrasts of the HCP dataset are presented in Figs 5 – Fig 11: *foot* contrast of the HCP motor task in Fig 5, *hand* contrast of the HCP motor task in Fig 6, *relational versus match* contrast of the HCP relational task in Fig 7, *gain vs loss* contrast of the HCP gambling task in Fig 8, *2-back vs 0-back* contrast of the HCP working memory task in Fig 9, *face vs shape*

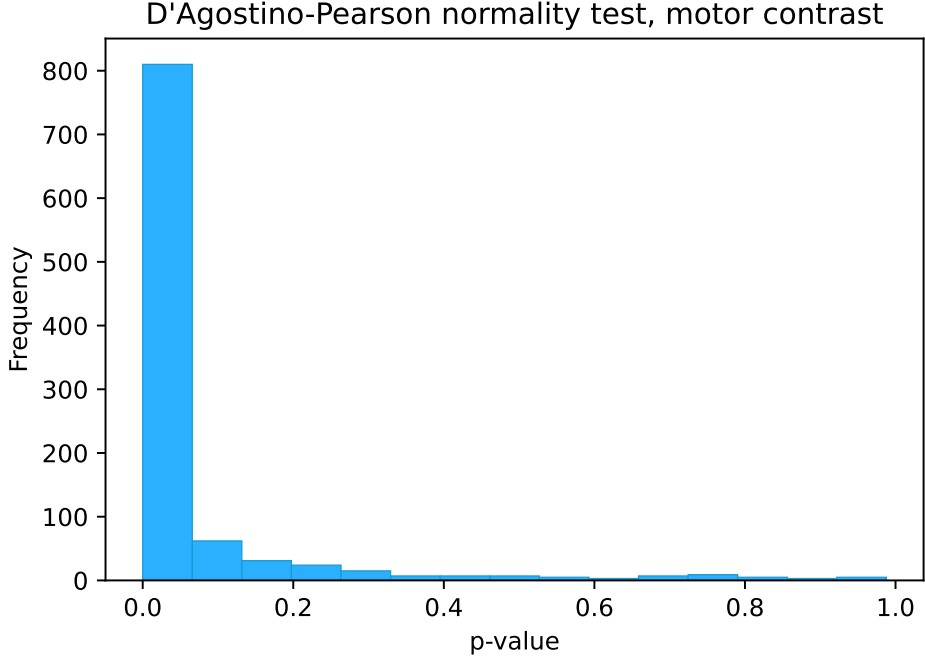

Figure 4: **D'Agostino-Pearson normality test for a motor contrast.** We perform a normality test for each cluster amongst the 1000 present for the motor foot contrast. The distribution of the normality test $p$-values indicates strong non-normality in fMRI data.

contrast of the HCP Emotional task in Fig 10, *interacting vs non-interacting* contrast of the HCP social task in Fig 11. These maps display the support of the conditional association test, with a sign that shows whether a region has an upward or downward impact on the decision function.

Overall, many Knockoff-based methods are powerless on all contrasts considered. Only KOPI, Vanilla Knockoffs and e-values aggregation consistently display non trivial solutions. This corresponds to the behavior observed in hard simulation settings in Fig. 1, i.e. low SNR and high correlation for instance.

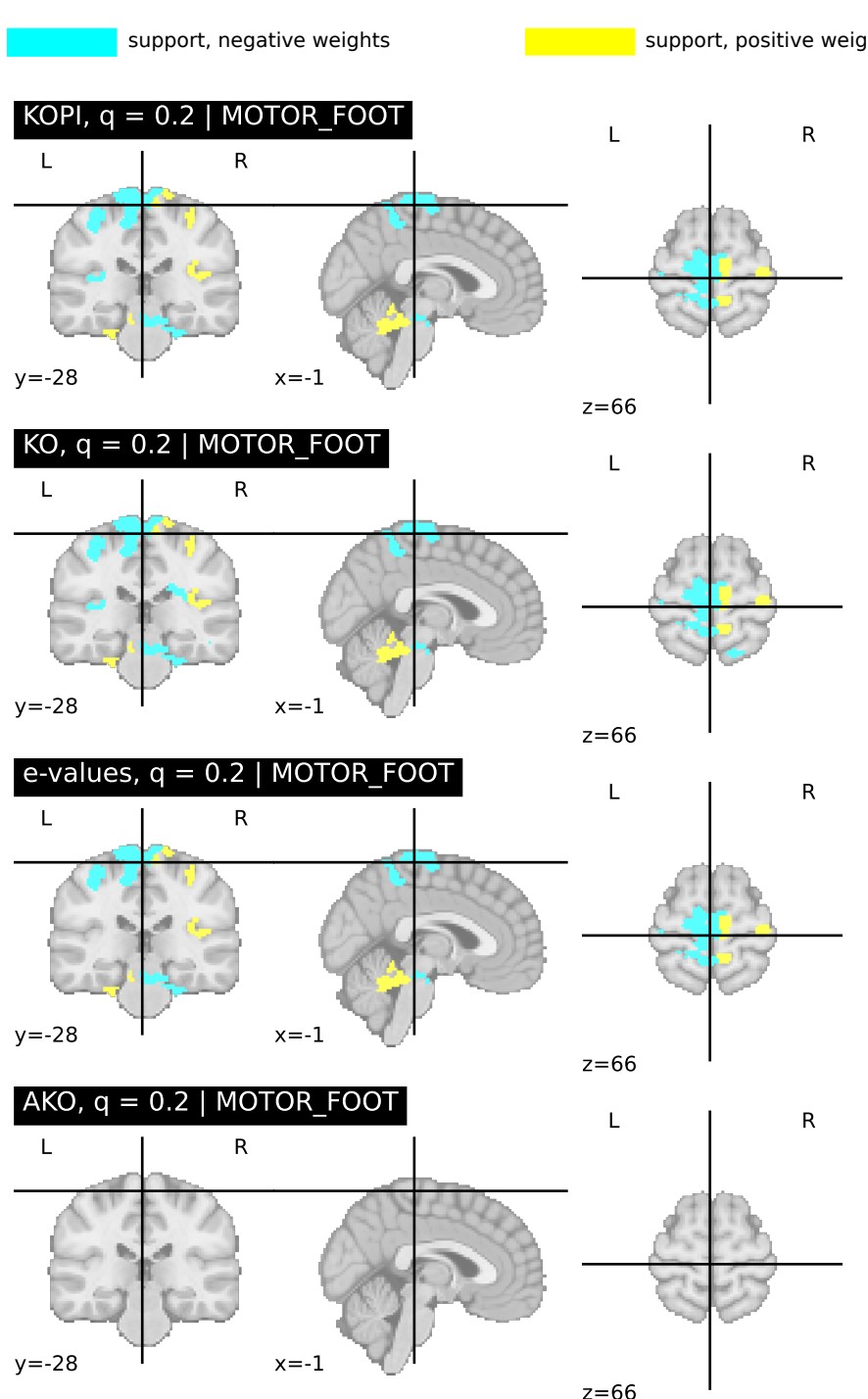

Figure 5: **Brain mapping on motor contrast using Knockoffs-based methods.** Among the five methods considered in this paper –Vanilla Knockoffs, aggregated Knockoffs using e-values, aggregated Knockoffs using quantile-aggregation (AKO), KOPI and Knockoff inference via Closed Testing– only Vanilla Knockoffs, e-values and KOPI yield discoveries, plotted above. All other methods are powerless. We use 50 Knockoffs draws and $\alpha = 0.1$ and $q = 0.2$. Each figure represents the region returned by a given method. Vanilla Knockoffs yield 17 regions, KOPI: 24 regions and e-values: 18 regions.

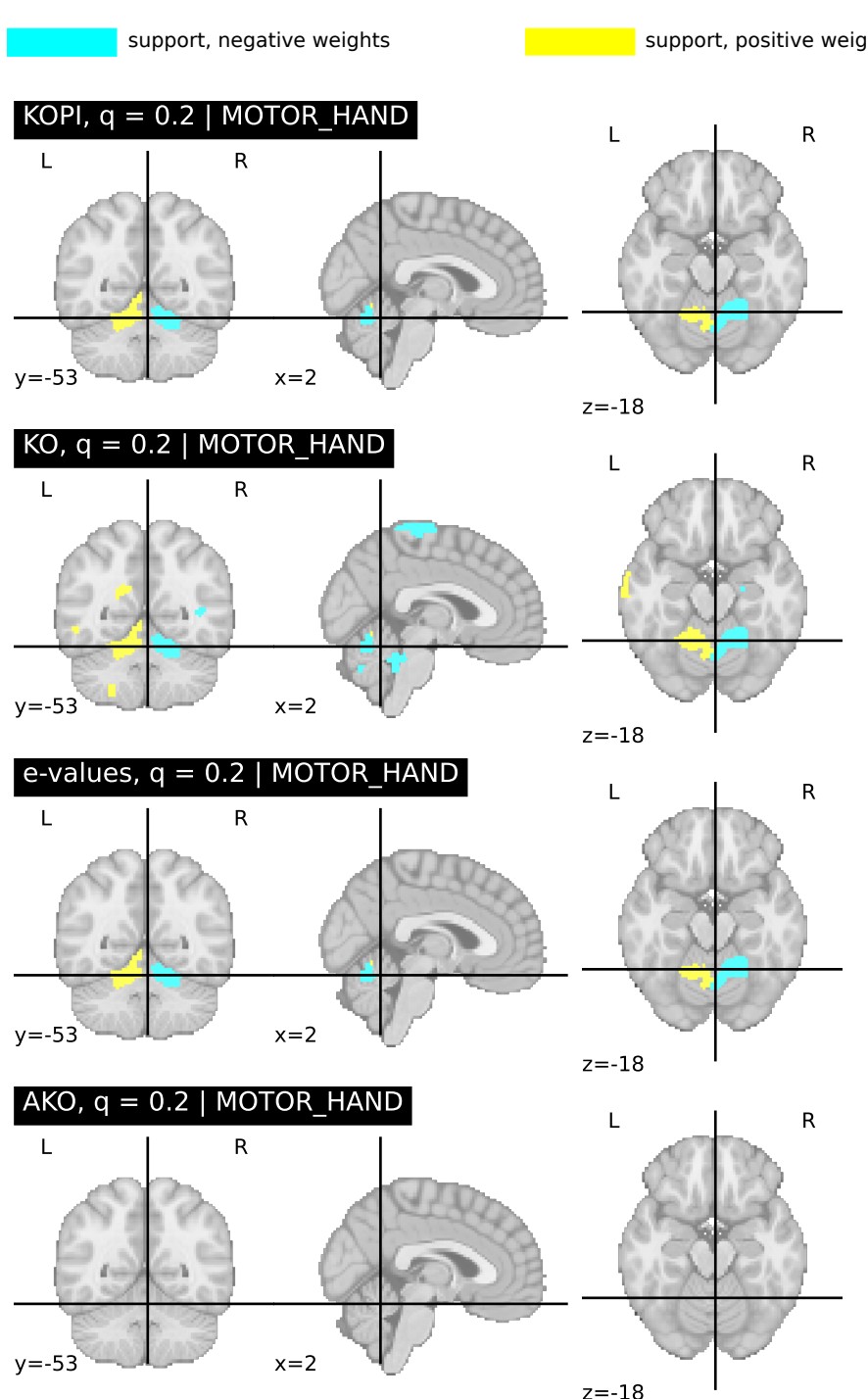

Figure 6: **Brain mapping on motor hand contrast using Knockoffs-based methods.** Among the five methods considered in this paper –Vanilla Knockoffs, aggregated Knockoffs using e-values, aggregated Knockoffs using quantile-aggregation (AKO), KOPI and Knockoff inference via Closed Testing– only Vanilla Knockoffs, e-values and KOPI yield discoveries, plotted above. All other methods are powerless. We use 50 Knockoffs draws and $\alpha = 0.1$ and $q = 0.2$. Each figure represents the region returned by a given method. Vanilla Knockoffs yield 11 regions, KOPI, 10 regions and e-values 11 regions.

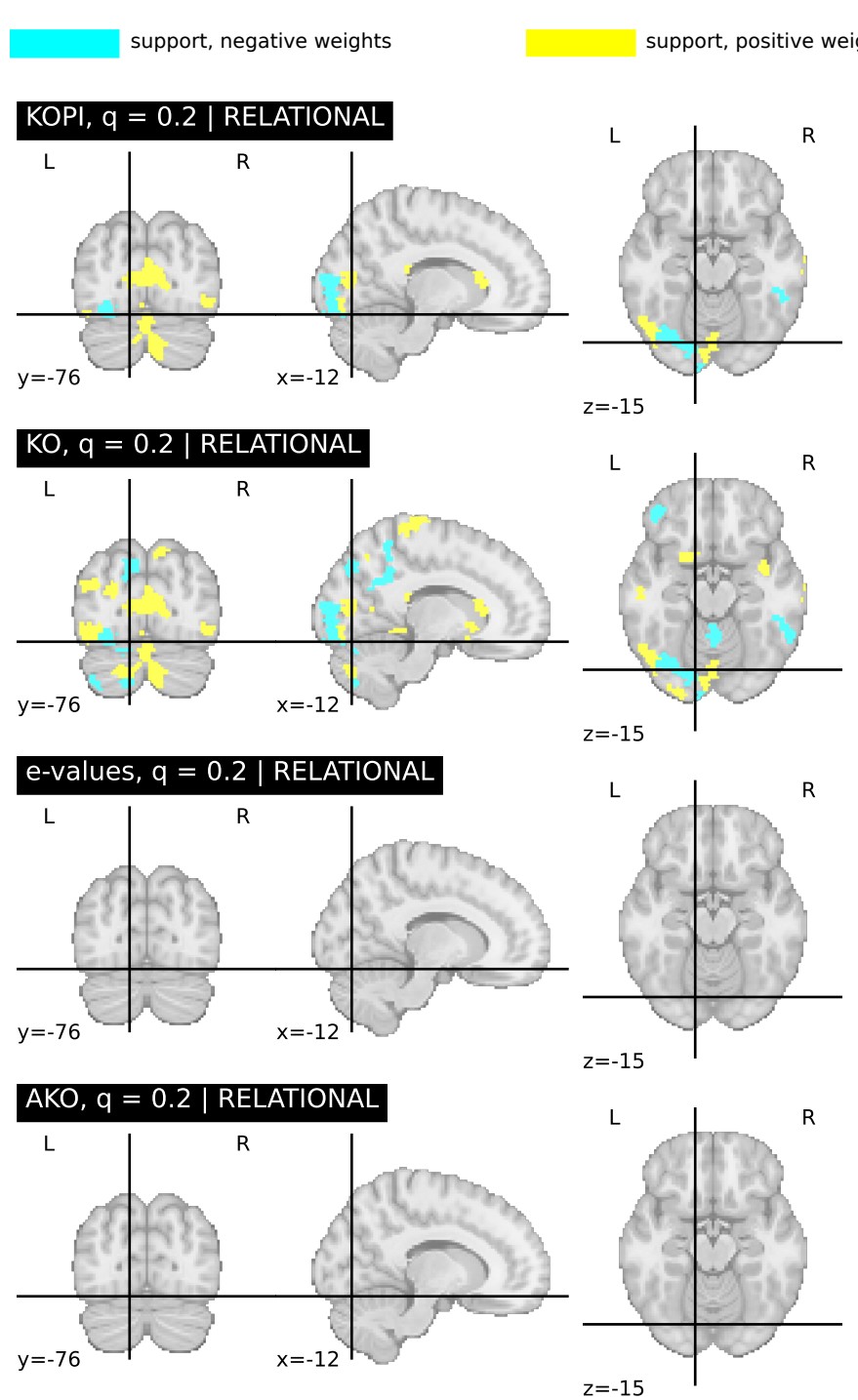

support, negative weights    support, positive weights

Figure 7: **Brain mapping on the HCP Relational task using Knockoffs-based methods.** Among the five methods considered in this paper –Vanilla Knockoffs, aggregated Knockoffs using e-values, aggregated Knockoffs using quantile-aggregation (AKO), KOPI and Knockoff inference via Closed Testing– only Vanilla Knockoffs and KOPI yield discoveries, plotted above. All other methods are powerless. We use 50 Knockoffs draws, $\alpha = 0.1$ and $q = 0.2$. Each figure represents the region returned by a given method. Vanilla Knockoffs yield 58 regions and KOPI, 24 regions.

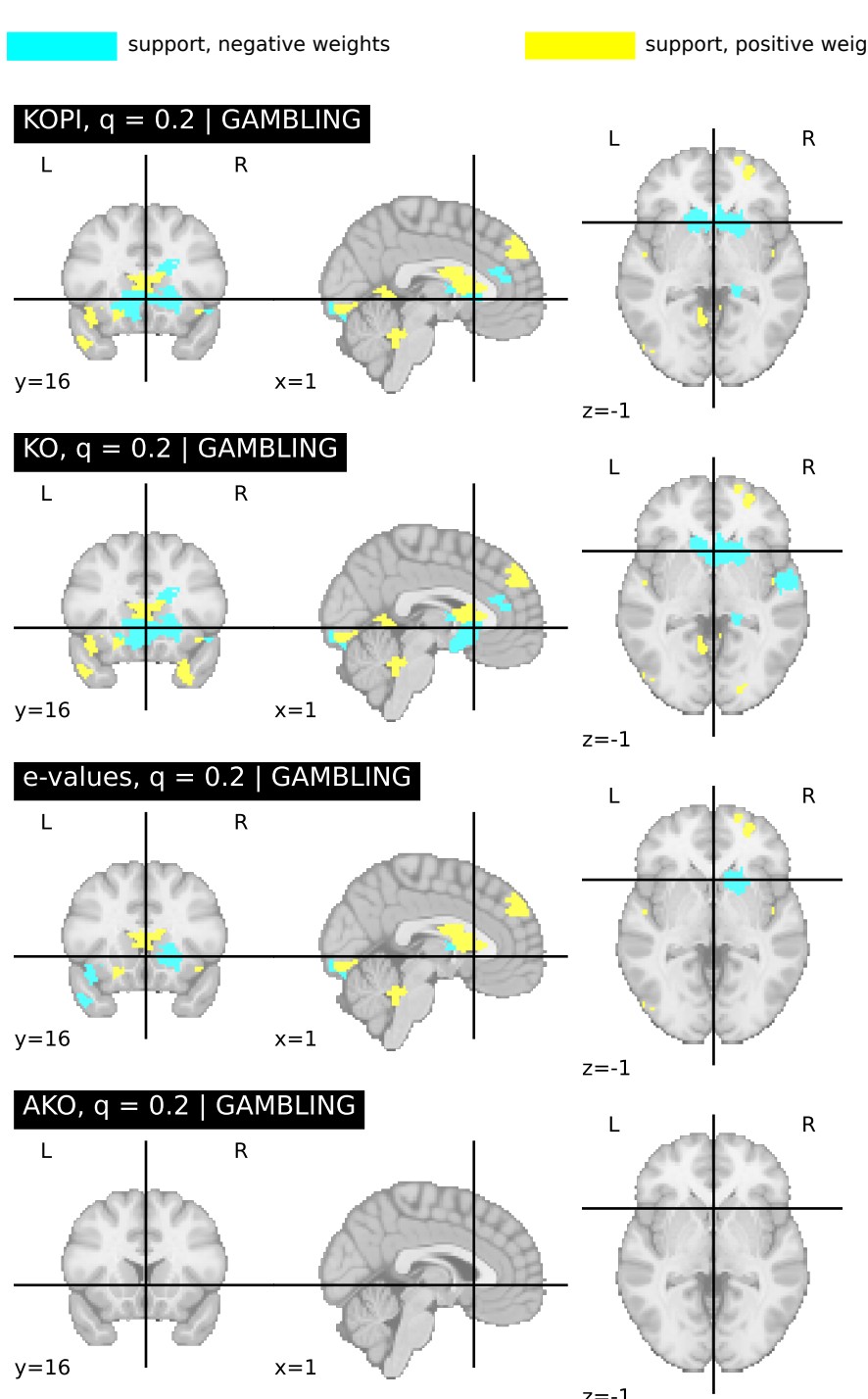

Figure 8: **Brain mapping on HCP gambling task using Knockoffs-based methods.** Among the five methods considered in this paper –Vanilla Knockoffs, aggregated Knockoffs using e-values, aggregated Knockoffs using quantile-aggregation (AKO), KOPI and Knockoff inference via Closed Testing– only Vanilla Knockoffs, KOPI and e-values aggregation yield discoveries, plotted above. All other methods are powerless. We use 50 Knockoffs draws, $\alpha = 0.1$ and $q = 0.2$. Each figure represents the region returned by a given method. Vanilla Knockoffs yield 57 regions, KOPI 57 regions, e-values aggregation, 19 regions.

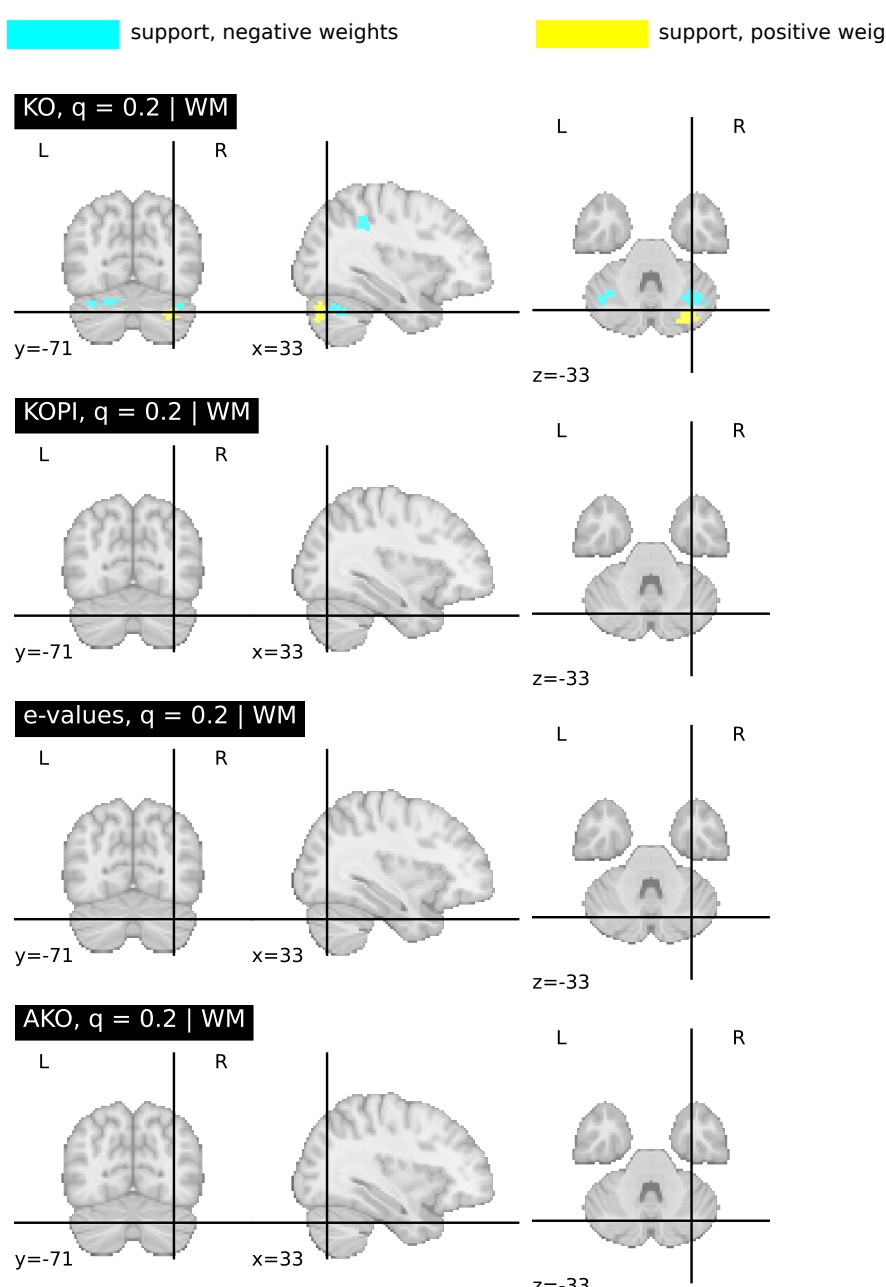

Figure 9: **Brain mapping on HCP working memory task using Knockoffs-based methods.**
Among the five methods considered in this paper –Vanilla Knockoffs, aggregated Knockoffs using
e-values, aggregated Knockoffs using quantile-aggregation (AKO), KOPI and Knockoff inference
via Closed Testing– only Vanilla Knockoffs yields discoveries, plotted above. All other methods are
powerless. We use 50 Knockoffs draws, $\alpha = 0.1$ and $q = 0.2$. Each figure represents the region
returned by a given method. Vanilla Knockoffs yield 8 regions.

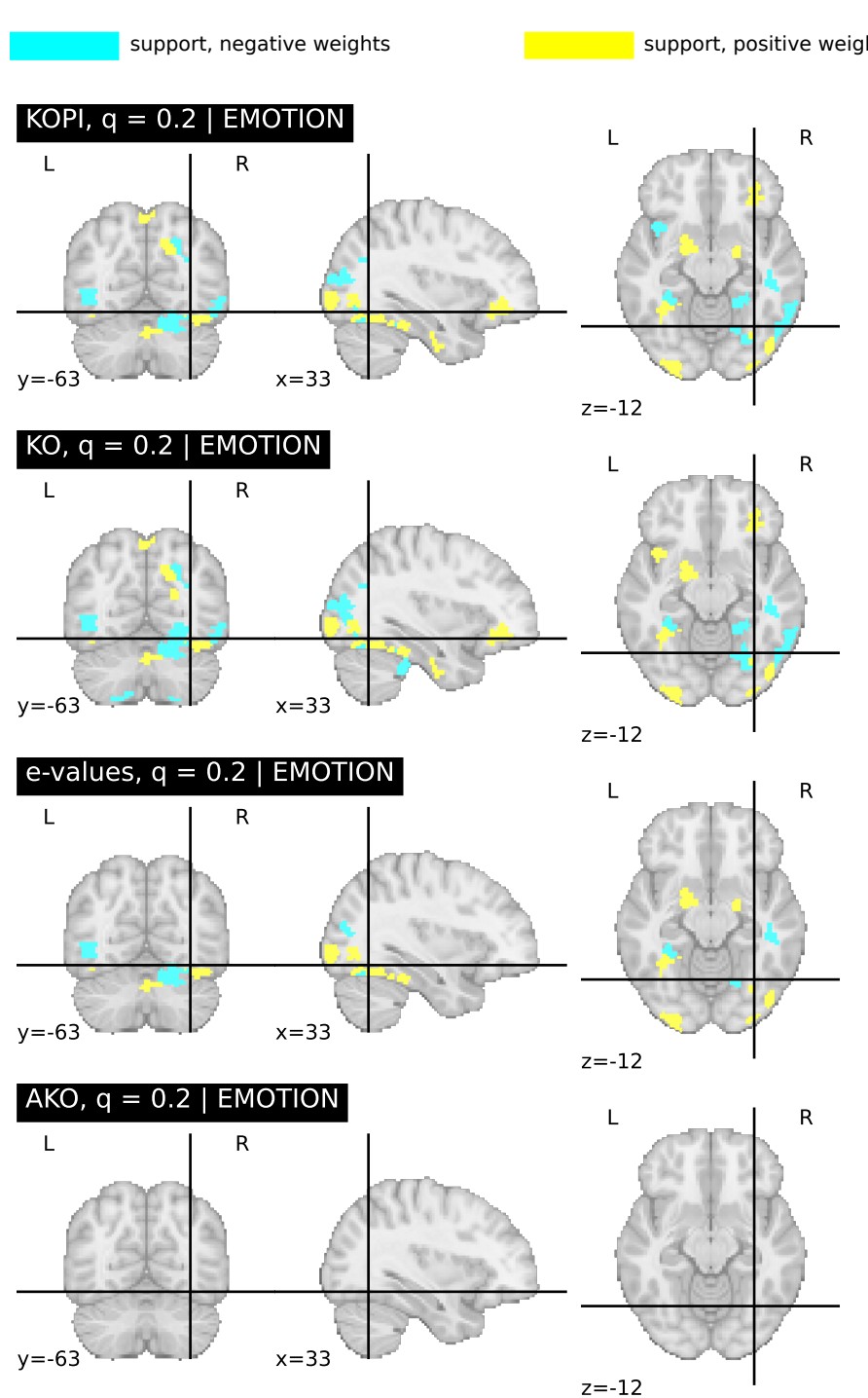

Figure 10: **Brain mapping on HCP emotional task using Knockoffs-based methods.** Among the five methods considered in this paper –Vanilla Knockoffs, aggregated Knockoffs using e-values, aggregated Knockoffs using quantile-aggregation (AKO), KOPI and Knockoff inference via Closed Testing– only Vanilla Knockoffs, KOPI and e-values aggregation yield discoveries, plotted above. All other methods are powerless. We use 50 Knockoffs draws, $\alpha = 0.1$ and $q = 0.2$. Each figure represents the region returned by a given method. Vanilla Knockoffs yield 22 regions, KOPI: 37 regions, e-values aggregation: 20 regions.

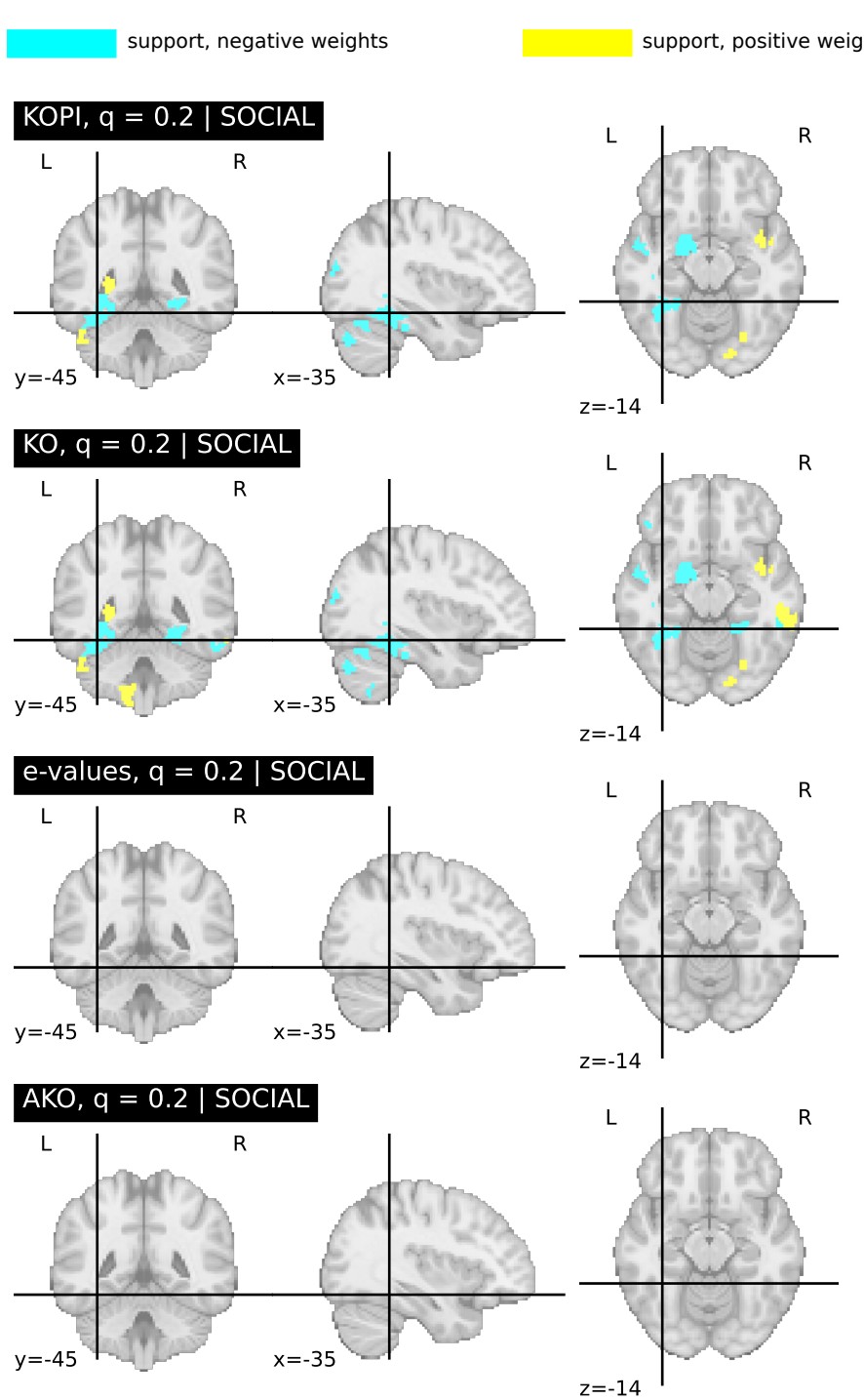

support, negative weights      support, positive weights

Figure 11: **Brain mapping on HCP social task using Knockoffs-based methods.** Among the five methods considered in this paper –Vanilla Knockoffs, aggregated Knockoffs using e-values, aggregated Knockoffs using quantile-aggregation (AKO), KOPI and Knockoff inference via Closed Testing– only Vanilla Knockoffs and KOPI yield discoveries, plotted above. All other methods are powerless. We use 50 Knockoffs draws, $\alpha = 0.1$ and $q = 0.2$. Each figure represents the region returned by a given method. Vanilla Knockoffs yield 32 regions, KOPI: 27 regions.