# OpenReview forum: "False Discovery Proportion control for aggregated Knockoffs"
_NeurIPS.cc/2023/Conference — NeurIPS 2023 poster_

### Official Review · Reviewer_DefH · 2023-06-23

**Soundness:** 3 good
**Presentation:** 3 good
**Contribution:** 3 good
**Rating:** 7
**Confidence:** 3

**Summary:**

This paper presents a method KOPI, based on the knockoff framework that controls the false discovery _proportion_ instead of the false discovery _rate_.  The key idea of the approach is to note that certain summaries of the knockoff statistics are exactly equal in distribution to certain functions of independent Rademacher random variables under the case of all nulls.  One can then estimate the FDP via Monte Carlo using draws of Rademachers, allowing for the setting of thresholds to obtain a desired FDP.  This approach is then straightforwardly extended to the case of aggregating multiple draws of the knockoffs.  The approach is shown to truly control the FDP while maintaining good power on some simulated datasets.

**Strengths:**

* The approach is clever and relatively straightforward.
* The approach does not rely on any distributional assumptions beyond the ability to generate knockoffs with the usual exchangeability property.
* The paper does a good job of introducing the knockoff framework and situating the present developments in that context.
* The approach seems well powered and has attractive theoretical guarantees.

**Weaknesses:**

* One minor weakness of the paper is that there are a number of potential tweaks to the method that are not explored or presented.  While the authors seem to have found choices that appear to work well, they appear somewhat arbitrary without additional context.  For example,   the choice of using harmonic mean for aggregation seems somewhat arbitrary without comparing to other choices, particularly the obvious choices like arithmetic or geometric mean. Another example is the choice of template.  The theoretical results are quite general, but then only a particular template is considered.

Typos:
* Line 75: is a word missing here: "knockoffs-based inference arbitrary aggregation schemes"?
* Line 76: "and increases sensitivity" --> "and increased sensitivity"
* Line 92: "complementary" --> "complement"
* Line 93: "cardinal" --> "cardinality"
* Line 140: Proposition 2 is empty and presumably included by mistake.
* Line 153: The equations following this line are assuming that $W_j$ is positive, and that should be stated.  E.g., if $W_j = -2$ and $W_k = 1$, then $1 = W_k \le -W_j = 2$ but $W_k$ is obviously not smaller than $0$.
* Line 154: This line is assuming that $W_1,\ldots, W_P$ are ordered, which they are not according to the notation. Either there should be a "without loss of generality, assume that $W$ is sorted such that $\sigma(W)$ is the identity" or all of the indices in the proof should really be $\sigma_j$ and $\sigma_k$ instead of $j$ and $k$.
* Line 184: "is arbitrarily close" should be changed to something like "can be made arbitrarily close by taking $B$ large enough"
* Line 191: "consists in finding" --> "consists of finding"
* Line 280: "face validity" --> "validity"
* Line 308: Saying that the method requires no assumption on the "law of the Knock statistics under the null" is ignoring that the Knockoffs much be exchangeable with the original data, no?

**Questions:**

* The proof of Theorem 1 essentially bounds $p_0$ by $p$ at one point.  Would it be possible to do the usual trick from FDR-type procedures of using a conservative estimate of $p_0$ in order to obtain a more powerful procedure?  This is not necessary for the present paper, but I'm curious how difficult such an approach would be.

**Limitations:**

I do not believe the present work has any potential negative societal impact.

---

> ### Author Rebuttal · Authors · 2023-08-07
>
> We thank the reviewer for their time and insightful comments. We also thank the reviewer for the typos found: they have been fixed in the manuscript. Please find our answers to the points raised below.
>
>
> > One minor weakness of the paper is that there are a number of potential tweaks to the method that are not explored or presented. While the authors seem to have found choices that appear to work well, they appear somewhat arbitrary without additional context. For example, the choice of using harmonic mean for aggregation seems somewhat arbitrary without comparing to other choices, particularly the obvious choices like arithmetic or geometric mean. Another example is the choice of template. The theoretical results are quite general, but then only a particular template is considered.
>
> While the theoretical guarantees we obtain hold for all choices of aggregation schemes and templates, these two hyperparameters indeed impact the power of KOPI. To address the reviewer's concern, we have performed an **additional experiment** on simulated data to compare four aggregation schemes: **arithmetic mean, geometric mean, harmonic mean and quantile aggregation [1].**
>
> We use the setup described in the main text - i.e. $\alpha = 0.1,  q = 0.1$ - and run the simulation 50 times. Importantly, we first check that the FDP is controlled for all types of aggregation and in all settings considered by reporting the bound non-coverage as described in the main text. We use three settings of varying difficulty, parametrized by the correlation level $\rho$:
>
> |              | Harmonic | Arithmetic | Geometric | Quantile aggregation |
> |--------------|----------|------------|-----------|----------------------|
> | $\rho = 0.5$ | $10$%    | $0$%       | $2$%      | $10$%                |
> | $\rho = 0.6$ | $2$%     | $0$%       | $0$%      | $4$%                 |
> | $\rho = 0.7$ | $2$%     | $0$%       | $0$%      | $0$%                 |
>
> The FDP is indeed controlled in all cases since non-coverage never exceeds the chosen level $\alpha = 10$%. **This is coherent with the theoretical guarantees we obtain in Theorem 3.** We now report the **average power** to benchmark aggregation schemes:
>
> |              | Harmonic | Arithmetic | Geometric | Quantile aggregation |
> |--------------|----------|------------|-----------|----------------------|
> | $\rho = 0.5$  | **0.91** | 0.77          | 0.87                      | 0.90   |
> | $\rho = 0.6$  | **0.83**     | 0.58        | 0.77        | **0.83**                |
> | $\rho = 0.7$ | **0.72**  | 0.39        | 0.61       | **0.72**                |
>
> Note that harmonic mean aggregation **outperforms arithmetic and gemeotric mean consistently** and performs similarly to quantile aggregation.
>
> Regarding the choice of template, we also considered using a linear template - i.e. $\mathbf{t}_k = \frac{\lambda k}{m}$ with $\lambda$ to choose by calibration. We found that using a nonparametric template that mimicks the shape of the $\pi$-statitistics under the null **consistently outperforms the linear template.** This is coherent with findings of [2].
>
> Besides aggregation and template choice, we have not introduced additional hyperparameters and we make a canonical use of knockoffs.
>
> >The proof of Theorem 1 essentially bounds $p$ by $p_0$ at one point. Would it be possible to do the usual trick from FDR-type procedures of using a conservative estimate of $p_0$ in order to obtain a more powerful procedure? This is not necessary for the present paper, but I'm curious how difficult such an approach would be.
>
> This is an interesting idea which seems related to the step-down procedure proposed in Algorithm 4.1 of [4]. Following this idea, we have to compute a conservative estimate of the null set $\mathcal{H}_0$ denoted $\widehat{\mathcal{H}_0}$ with $|\widehat{\mathcal{H}_0}| = \widehat{p_0}$ via vanilla KOPI and use it to sharpen the bound obtained in Theorem 1. We believe that the implicit bounding of $p$ by $p_0$ that the reviewer describes happens at this step of the proof:
>
> >In the case where $\mathcal{H}_0 \subsetneq [[p]]$, false null $\chi_j$ will insert $-1$'s into the process on the nulls, implying that $N_k$ is stochastically dominated by $N^0_k$
>
> Concretely, sharpening this bound requires using: \begin{align}
> \widehat{N^0_k} =  \left| \{
>       j \in \widehat{\mathcal{H}_0}, \chi^0_j=1 \textrm{ and } \frac{1 + Z^0_j}{p} < t_k
>      \}\right|
>   \end{align}
>
> with the notation defined above. However, we are not sure what theoretical guarantees can be obtained using $\widehat{N^0_k}$ instead of $N^0_k$. The problem that arises is that **$\widehat{N^0_k}$ depends on the data** and therefore the rest of the proof does not hold.
>
>
> This discussion about a step-down version has been added to the paper.
>
>
> **References**
>
> [1] Meinshausen, N., Meier, L., & Bühlmann, P. (2009). P-values for high-dimensional regression. Journal of the American Statistical Association, 104(488), 1671-1681.
>
> [2] Blain, A., Thirion, B., & Neuvial, P. (2022). Notip: Non-parametric True Discovery Proportion control for brain imaging. NeuroImage, 260, 119492.
>
> [4] Blanchard, G., Neuvial, P., & Roquain, E. (2020). Post hoc confidence bounds on false positives using reference families. Annals of Statistics.

---

> > ### Comment · Reviewer_DefH · 2023-08-10
> >
> > Thank you for the thorough response to my comments.  Hopefully they were helpful -- I believe that the presented results strengthen the claims in the paper.

---

> > > ### Author Response · Authors · 2023-08-14
> > >
> > > We thank the reviewer for their prompt response and again for their helpful comments. We remain at the reviewer's disposal in case any additional clarification is needed.

---

### Official Review · Reviewer_zfG6 · 2023-07-03

**Soundness:** 2 fair
**Presentation:** 2 fair
**Contribution:** 2 fair
**Rating:** 5
**Confidence:** 1

**Summary:**

I can understand this paper but I don't have sufficient knowledge to make a solid judgment on it, please ignore my review.

**Strengths:**

I can understand this paper but I don't have sufficient knowledge to make a solid judgment on it, please ignore my review.

**Weaknesses:**

I can understand this paper but I don't have sufficient knowledge to make a solid judgment on it, please ignore my review.

**Questions:**

I can understand this paper but I don't have sufficient knowledge to make a solid judgment on it, please ignore my review.

**Limitations:**

I can understand this paper but I don't have sufficient knowledge to make a solid judgment on it, please ignore my review.

---

### Official Review · Reviewer_DZHu · 2023-07-04

**Soundness:** 3 good
**Presentation:** 3 good
**Contribution:** 3 good
**Rating:** 6
**Confidence:** 4

**Summary:**

This paper studies an important topic: variable selection. Concretely, the authors proposed a novel method KOPI which can theoretically control false discovery proportion at a pre-specified level. Besides, the authors also conduct lots of experiments based on simulated data and real data to verify the effectiveness of their proposed procedure.

**Strengths:**

1，	This paper is well-written, its notations and definitions are clear;

2，	The authors provide some necessary theoretical guarantees for their proposed algorithm.


**Weaknesses:**

1，	Simulation experiments and real-data experiments are conducted, However, in order to verify the effectiveness of KOPI, more experiments on other real-data datasets are needed;

2，	It will be better if the authors can provide an analysis of the false nondiscovery proportion (FNP), a dual quantity of FNP, or its expectation FNR., see [1] .

[1] Genovese C, Wasserman L. Operating characteristics and extensions of the false discovery rate procedure[J]. Journal of the Royal Statistical Society Series B: Statistical Methodology, 2002, 64(3): 499-517.

**Questions:**

No

---

> ### Author Rebuttal · Authors · 2023-08-07
>
> We thank the reviewer for their time and insightful comments. Please find our answers to the points raised below.
>
> > Simulation experiments and real-data experiments are conducted, However, in order to verify the effectiveness of KOPI, more experiments on other real-data datasets are needed;
>
> To address the reviewer's remark regarding real-world datasets, we performed an additional experiment on genomics data. The dataset we used is part of **GEMLeR (Gene Expression Machine Learning Repository)** [2] a collection of gene expression datasets that can be used to benchmark machine learning methods on genomics data.
>
> We chose the "Colon vs Kidney" dataset: this is a binary classifcation dataset where the goal is to distinguish cancerous tissue from two different organs (Colon and Kidney) using gene expression data. This dataset comprises $546$ samples and $10936$ genes. To make the problem tractable for Knockoffs-based methods we perform dimensionality reduction to select the $546$ genes that have the largest variance. Then, **we run all Knockoffs-based methods 50 times** and report the selected genes.
>
> |                            | KOPI   | Vanilla KO | e-values | Closed Testing | AKO |
> | -------------------------- | ------ | ---------- | -------- | -------------- | --- |
> | Selected in > 90% of runs  | **21** | 0          | 0        | 0              | 0   |
> | Selected in > 50% of runs  | 22     | 25         | 0        | 0              | 0   |
> | Spurious detections (<50%) | **7**  | 34         | 20       | 0              | 0   |
>
> We display **stability selection criteria** for 5 Knockoffs-based methods. Note that KOPI displays a **very stable selection set across all runs** with $21$ genes present in $>90$% of runs. **KOPI also avoids most spurious discoveries**, as only $7$ genes are selected less than $50$% of the time, compared to $34$ genes using Vanilla Knockoffs and $20$ using $e$-values aggregation. All other Knockoffs-based methods are powerless in all runs.
>
> This experiment further shows that KOPI outperforms all other Knockoffs-based methods **on real-world data** in terms of selection stability.
>
>
> > It will be better if the authors can provide an analysis of the false nondiscovery proportion (FNP), a dual quantity of FNP, or its expectation FNR., see [1].
>
> Analyzing the FNP of KOPI is a very interesting perspective. However, it is challenging for several reasons. A first technical difficulty is that the analysis of the FNP made in [1] relies **on the assumption that the test statistics are independent, uniformly distributed under the null hypothesis, and identically distributed under the alternative hypothesis.** In this line of work, a first asymptotic analysis of the FNP/Power of JER controlling procedures is made in section 6.2 of [4].
>
> This requires making relatively strong assumptions about the alternative distribution. An important contribution of the present paper is that **we are not assuming independence between the test statistics or assuming their identical distribution under the alternative.** In contrast, we show that the joint null distribution of the $\pi$ statistics can be sampled from, and our approach is agnostic with respect to their distribution under the alternative hypothesis.
>
> Furthermore, the FNP may be difficult to interpret in the context of Knockoffs for various reasons. Indeed, **the fiability of Knockoff statistics may be undermined when input variables are highly correlated [3].** In short, relevant variables that are highly correlated with irrelevant variables may be assigned a negative statistic. This leads to False Negatives for all Knockoffs-based methods. Also, **the FNP is affected by the sparsity of the signal:** Knockoffs-based methods are unable to make rejections if there are too few large positive Knockoff statistics.
>
> We have added a paragraph about FNP analysis to the Discussion section of the manuscript.
>
>
> **References**
>
> [1] Genovese C, Wasserman L. Operating characteristics and extensions of the false discovery rate procedure[J]. Journal of the Royal Statistical Society Series B: Statistical Methodology, 2002, 64(3): 499-517.
>
> [2] Stiglic, G., & Kokol, P. (2010). Stability of ranked gene lists in large microarray analysis studies. Journal of biomedicine and biotechnology, 2010.
>
> [3] Spector, A., & Fithian, W. (2022). Asymptotically Optimal Knockoff Statistics via the Masked Likelihood Ratio. arXiv preprint arXiv:2212.08766.
>
> [4] Blanchard, G., Neuvial, P., & Roquain, E. (2020). Post hoc confidence bounds on false positives using reference families. Annals of Statistics.

---

### Official Review · Reviewer_MbEh · 2023-07-06

**Soundness:** 3 good
**Presentation:** 3 good
**Contribution:** 3 good
**Rating:** 6
**Confidence:** 2

**Summary:**

In this paper, the authors discuss controls for false discoveries in variable selection. While how to perform FDR control is known, authors opt to control FDP, which is a random quantity depending on a specific dataset. The main concept is an upperbound of FDP (proposition 1), namely JER. The monte carlo version of JER is applied to perform FDP control. The authors also propose an aggregated JER to reduce the stochasticity JER for variable selection. Experiments show good FDP control while maintaining good power.

**Strengths:**

1. The paper is very well written.
2. FDP (not just FDR) control is an important problem. This paper targets on an important issue in our community.
3. The proposed method has clear theoretical support (theorem 2 and 3), although I have some questions.

**Weaknesses:**

It is not entirely clear to me the delta between this paper and an important related work, i.e, [4]. From Section 4, it seems that using JER to control FDP has already been proposed. Thus, I wonder how significant the contribution of this work is.

For example, has Proposition 1 or a similar version already been proposed in [4]? I would appreciate a short paragraph comparing this work and [4].

**Questions:**

I am not sure Why JER, not the empirical JER is studied in Theorem 2? In practice, shouldn't one use empirical JER with monte carlo samples?

**Limitations:**

Yes.

---

> ### Author Rebuttal · Authors · 2023-08-07
>
> We thank the reviewer for their time and insightful comments. Please find our answers to the points raised below.
> > It is not entirely clear to me the delta between this paper and an important related work, i.e, [4]. From Section 4, it seems that using JER to control FDP has already been proposed. Thus, I wonder how significant the contribution of this work is.
>
>
> >For example, has Proposition 1 or a similar version already been proposed in [4]? I would appreciate a short paragraph comparing this work and [4].
>
> While the JER framework has indeed been proposed in [4], this work's main contributions differ substantially from [4]:
>
> * [4] focuses on marginal testing (e.g. massively multiple t-tests) and not conditional testing. KOPI is the first procedure to combine the JER framework and **conditional testing via the Knockoffs framework and $\pi$-statistics.**
>
> * In general, the JER cannot be computed analytically since it requires knowledge of the distribution of the test statistics under the null. In practice [4] relies on permutation schemes to estimate this distribution. This is computationally infeasible in the context of Knockoffs: such a procedure would require computing new Knockoffs for each permutation run. Theorem 1 of this work is, to our knowledge, **the first tractable upper bound on the JER of the $\pi$-statistics or any other Knockoffs-based statistic**.
>
> * **KOPI supports aggregation**. Aggregation of test statistics is not considered in [4], which focuses on JER control on $p$-values. In this work, we show that **FDP control stemming from JER control can be made robust using aggregation.**
>
> Regarding Proposition 1: indeed this proposition corresponds to Proposition 2.3 in [4] - we simply restate it for completeness and self-containedness and include the proof in the supplementary material. To clarify this, **we have added a sentence in the manuscript** that points to the proposition in [4].
>
> > I am not sure Why JER, not the empirical JER is studied in Theorem 2? In practice, shouldn't one use empirical JER with monte carlo samples?
>
> To reach exact FDP control, the **JER itself** - and not the empirical JER - has to be controlled. While indeed $\mathbf{t}^B_\alpha$ is built using Monte Carlo samples in practice and controls the empirical JER by definition, we need to insure that **the JER of $\mathbf{t}^B_\alpha$ is controlled to obtain FDP control**. Theorem 2 shows that this is indeed the case up to Monte Carlo error, which can be made arbitrarily small.
>
> **References**
>
> [4] Blanchard, G., Neuvial, P., & Roquain, E. (2020). Post hoc confidence bounds on false positives using reference families. Annals of Statistics.

---

> > ### Comment · Reviewer_MbEh · 2023-08-14
> > **Thanks for replying!**
> >
> > >  Theorem 2 shows that this is indeed the case up to Monte Carlo error, which can be made arbitrarily small.
> >
> > Yep, this clarifies my confusion.
> >
> > > this work's main contributions differ substantially from [4]: ...
> >
> > This also clarifies my confusion. Thus I suggest authors include a version of these clarifications in the revision.
> >
> > I continue to lean toward accepting this paper as I believe FDP bounding is an important issue and the tractable upperbound on JER is interesting. Based on the novelty claims authors made, I will raise my score by 1.

---

> > > ### Author Response · Authors · 2023-08-15
> > >
> > > We thank the reviewer for their prompt response and careful consideration of our answers. We appreciate their decision to raise their score and remain at the reviewer's disposal in case any additional clarification is needed.

---

### Author Rebuttal · Authors · 2023-08-07

Rebuttal Summary
----

We thank all three reviewers for their time and comments. Here is a summary of the elements we addressed in our answers:

* We have added **two experiments** to address the reviewers' concerns on **real-world datasets** and the choice of the aggregation scheme. We use a gene expression dataset and show that **KOPI improves selection set stability** compared to other Knockoffs-based methods while avoiding spurious discoveries. In the second experiment, we benchmark harmonic mean aggregation against other alternatives on simulated data.

* We have clarified the distinction between this work's main contribution and the paper which introduces FDP control via JER control [4] for marginal testing. In short, the present work is **the first procedure to combine the JER framework and conditional testing via the Knockoffs framework and $\pi$-statistics.** The main contribution of the paper is obtaining a **tractable upper bound on the JER of the Knockoffs $\pi$-statistics.** This leads to FDP control in the Knockoffs framework, which hadn't been achieved previously. Furthermore, the procedure we propose supports **aggregation of test statistics to robustify inference,** which is not considered in [4].

* We have investigated interesting theoretical points raised by the reviewers, notably False Negative Proportion analysis and a possible step-down version of the proposed procedure.

We remain at the reviewers' disposal, should they have any additional questions or remarks.

**References**

[4] Blanchard, G., Neuvial, P., & Roquain, E. (2020). Post hoc confidence bounds on false positives using reference families. Annals of Statistics.

---

### Decision · Program_Chairs · 2023-09-21

**Decision:**

Accept (poster)

**Comment:**

This is a strong paper for which all three reviewers recommend acceptance.  Nevertheless, reviewers and authors had a robust discussion of many concerns and potential improvements.  Authors are strongly encouraged to incorporate their replies and the discussion into the final version of this paper.